# Multimodal Prescriptive Deep Learning

**Dimitris Bertsimas**                                          *dbertsim@mit.edu*
*Sloan School of Management and Operations Research Center, Massachusetts Institute of Technology, Cambridge, MA*

**Lisa Everest**                                          *leverest@mit.edu*
*Operations Research Center, Massachusetts Institute of Technology, Cambridge, MA*

**Vasiliki Stoumpou**                                          *vasstou@mit.edu*
*Operations Research Center, Massachusetts Institute of Technology, Cambridge, MA*

**Reviewed on OpenReview:** *https://openreview.net/forum?id=AwfWOCVLbJ*

## Abstract

We introduce a multimodal deep learning framework, Prescriptive Neural Networks (PNNs), that combines ideas from optimization and machine learning to perform treatment recommendation, and that, to the best of our knowledge, is among the first prescriptive approaches tested with both structured and unstructured data within a unified model. The PNN is a feedforward neural network trained on embeddings to output an outcome-optimizing prescription. In two real-world multimodal datasets, we demonstrate that PNNs prescribe treatments that are able to greatly improve estimated outcome rewards; by over 40% in transcatheter aortic valve replacement (TAVR) procedures and by 25% in liver trauma injuries. In four real-world, unimodal tabular datasets, we demonstrate that PNNs outperform or perform comparably to other well-known, state-of-the-art prescriptive models; importantly, on tabular datasets, we also recover interpretability through knowledge distillation, fitting interpretable Optimal Classification Tree models onto the PNN prescriptions as classification targets, which is critical for many real-world applications. Finally, we demonstrate that our multimodal PNN models achieve stability across randomized data splits comparable to other prescriptive methods and produce realistic prescriptions across the different datasets.

## 1 Introduction

Today's society provides an increasing availability of large quantities of data, particularly multimodal data consisting of structured and unstructured elements. As a result, developing systematic and personalized decision-making methods that can leverage such multimodal data becomes more and more critical, and the benefits of data-driven methods become more and more visible. For example, medical professionals could systematically and optimally treat patients based on individual characteristics, clinical notes, and medical scans (Soenksen et al., 2022). Companies in technology and digital advertising would be able to increase customer impact by customizing content and advertisements according to user-specific data. In the retail industry, such personalized models would allow companies to dynamically price goods and services based on the user or environment for increased revenue.

Much of the current work in machine learning and deep learning focuses on improving the accuracy of output prediction. We find that deep learning has tremendous and underutilized potential in the area of decision-making. This paper combines ideas from machine learning and optimization to move from prediction to prescription, with the ability to leverage multimodal data. We introduce a novel, multimodal, deep learning framework that we call a Prescriptive Neural Network (PNN). We demonstrate how our models handle complex data structures and how effective they are in both multimodal and unimodal real-world applications. Through these applications, we show that our PNN models are flexible with different treatment scenarios

that cover all real-life application settings. Our models also give stable and realistic results, comparable to existing prescriptive methods, and provide the user with more control over the resulting prescriptions. On tabular datasets in particular, we are able to recover interpretability by applying knowledge distillation and fitting interpretable Optimal Classification Tree (OCT) models (Bertsimas & Dunn, 2017; 2019) on the PNNs' prescriptions as classification targets; we find that these Mirrored OCTs perform comparably to their PNN counterparts, meaning that interpretability comes with little cost to performance.

## 1.1 Related Literature

Previous literature in data-driven personalized decision-making includes the Regress & Compare framework, tree-based methods, and causal methods.

**Regress & Compare.** Regress & Compare is a black-box methodology where a regression model is trained to predict the outcome under each treatment. The set of features used for training is the augmented feature data combined with the historical treatment given. Given an input observation and possible treatment options, the model is then used to select, for each sample, the treatment with the lowest (highest) outcome for a minimization (maximization) problem.

In this area, Zhao et al. (2012) introduces a framework that aims to estimate individual treatment rules. The goal is to assign treatments that maximize (or minimize) the expected outcome for each individual, by estimating the potential outcomes under each treatment (using an outcome weighted learning approach) and selecting the treatment that leads to the best outcome.

There are also many applications of the Regress & Compare methodology for prediction – examples include energy economics (Ferkingstad et al., 2011) and multidrug-resistant tuberculosis (Siddique et al., 2019). Other works use the Regress & Compare framework to move from predictions to prescriptions. In particular, Bertsimas & Kallus (2020) extends Regress & Compare solutions for prescriptive problems and incorporates $k$-nearest neighbors regression ($k$NN Altman (1992)), local linear regression (LOESS Cleveland & Devlin (1988)), classification and regression trees (CART Breiman et al. (1984)), and random forests (RF Breiman (2001)). Although Bertsimas & Kallus (2020) demonstrates that their methods are widely applicable and computationally tractable under mild conditions, we note that these are classical machine learning methods and do not take advantage of deep learning.

More specific applications of Regress & Compare for prescriptive problems include healthcare (Bertsimas et al., 2017a; Bayati et al., 2014) and revenue management (Bertsimas & Kallus, 2022). Bertsimas et al. (2017a) considers personalized diabetes treatment, while Bayati et al. (2014) combines prediction and decision-making to allocate interventions for post-discharge patients that were admitted due to heart failure. Bertsimas & Kallus (2022) considers the problem of optimal pricing, where they learn from historical observational data to optimize predicted revenue given price.

One possible limitation of the Regress & Compare approach is that it is affected by the number of samples per treatment, since for each treatment, only the samples that received that treatment in real life are considered. Also, it does not address the potential treatment assignment bias present in the data; e.g. healthier patients tend to receive lighter treatment and to have better outcomes. This is discussed in more detail in Section 2.3. Furthermore, the black-box nature reduces its interpretability, which is important for many real-world applications.

**Tree-based methods.** Kallus (2017) introduces Personalization Trees, which extend the Regress & Compare method for the problem of choosing the treatment with the best causal effect from a finite number of discrete options. Kallus presents three different recursive-partitioning-based algorithms: a greedy Personalization Tree, a Personalization Forest that bags Personalization Trees, and a globally optimal Personalization Tree. As these are tree methods, we note that they preserve interpretability.

Bertsimas et al. (2019) introduces Optimal Prescriptive Trees, which are similar to Personalization Trees but combine the counterfactual estimation and prescriptive learning tasks together in one training process and extend the framework of Optimal Classification Trees from Bertsimas & Dunn (2017; 2019). As a result, the trees are highly interpretable. Amram et al. (2022) further explores the optimal trees methodology and proposes Optimal Policy Trees, in which counterfactual estimation is performed separately from the

prescriptive learning task. This allows for greater flexibility in discrete and continuous treatments, as well as better learning of the prescriptive task due to reduced complexity that results from the separation of the two training tasks. Like Optimal Prescriptive Trees, this method also preserves interpretability. These approaches, however, struggle with learning more complicated functional forms and are therefore limited to learning outcome functions that can be modeled by trees of relatively small depth.

**Causal methods.** This family of methods originates from the causal inference literature and includes both individual trees (causal trees) and their combinations (causal forests). Athey & Imbens (2016) introduces causal trees, which employ a recursive partitioning approach of the feature space to split the data into groups with similar treatment effects. Causal forests extend causal trees and represent a prescriptive black-box method that builds on random forests, as introduced by Wager & Athey (2018). While random forests are constructed from decision trees, causal forests are composed of causal trees, which aim to maximize the difference in outcomes between two treatments at each node during tree growth. The resulting outcomes are interpreted relatively to one another. In the binary treatment case, since there are only two options (treatment or no treatment), one option will yield a positive effect (outcome) and the other a negative effect. If the goal is to minimize the outcome, the treatment option with the negative effect is prescribed.

Other models in the causal inference literature include causal boosting (Powers et al., 2018) and causal MARS (Powers et al., 2018). However, the estimation of treatment effects, which is achieved by causal models, is not an explicit policy prescription, which is the goal of this work.

Finally, another approach by Zhou et al. (2023) takes inspiration from the causal inference literature and uses inverse propensity weight estimators to calculate the counterfactuals. Decision trees (both greedy and fully optimal) are then used for policy learning. Fully optimal trees, however, struggle with scalability, while the heuristic-based trees do not guarantee the best possible policy (optimality).

**Deep learning methods.** Others have taken a deep learning approach to the optimal prescription problem. Patil et al. (2024) introduces prescriptive networks that are shallow neural networks to address the binary treatment regime, in which a treatment may or may not be given. Their networks are optimized by over-estimating conditional average treatment effects (CATE), and they propose a method using mixed-integer programming (MIP) to implement their networks into commercial solvers. Sun & Tsiourvas (2023) proposes a piecewise linear neural network model to output optimal prescriptions from a set of discrete treatments and show that their model partitions the input space into disjoint polyhedra, where all observations in the same partition are assigned the same treatment. Bergman et al. (2022) proposes a solver that takes as input user-specified pretrained predictive models (including neural networks) and formulates optimization models directly over those predictive models to provide final prescriptions.

Additionally, Shalit et al. (2017); Shi et al. (2019) lie at the intersection of causal methods and deep learning; they use neural networks to estimate causal effects. We briefly note that this is different from directly prescribing treatments to solve the optimal prescription problem. Shalit et al. (2017) proposes a general framework called Counterfactual Regression (CFR) and its variant, the Treatment-Agnostic Representation Network (TARNet). These models are designed to facilitate Individual Treatment Effect (ITE) estimation through a fully differentiable learning process that employs a regularized objective optimized via a deep feed-forward network consisting of six exponential-linear activation layers. Similarly, Shi et al. (2019) introduces Dragonnet, an alternative neural architecture tailored for ITE estimation. Its architecture is a three-headed structure that jointly models the propensity score and potential outcomes.

We note that, like our PNN models, all of these works combine ideas from optimization and machine learning. However, Bergman et al. (2022) does not incorporate Deep Learning in the prescriptive part of the framework, but only to generate predictions. Patil et al. (2024); Sun & Tsiourvas (2023) consider binary and discrete treatments respectively, while our work handles more treatment and outcome scenarios. Furthermore, our approach differs in the network's objective function used for training. Finally, Shalit et al. (2017); Shi et al. (2019) propose methods for causal estimation rather than direct prescription purposes, and though there is a possibility for extension to discrete treatments, they both only consider and report results for binary treatments.

### 1.2 Contributions

Our contributions are as follows:

1. Combining machine learning and optimization, we propose a novel, multimodal, deep learning framework we call a Prescriptive Neural Network (PNN); to the best of our knowledge, this is among the first prescriptive frameworks to handle multimodal data by integrating pretrained image and text embeddings with clinical tabular features.

2. In two real-world multimodal datasets, we demonstrate that PNNs prescribe treatments that are able to greatly improve estimated outcome rewards in transcatheter aortic valve replacement (TAVR) procedures (by over 40%) and in liver trauma injuries (by 25%). Additionally, PNNs either outperform or perform comparably to existing, state-of-the-art, prescriptive methods on four real-world unimodal (tabular) datasets that span all four treatment scenarios: diabetes management (multiple continuous treatments), groceries pricing (single continuous treatment), splenic injuries treatment (multiple discrete treatments), and REBOAs in blunt trauma patients (binary treatment).

3. On tabular datasets we recover interpretability through knowledge distillation; we train Optimal Classification Trees (OCT) (Bertsimas & Dunn, 2017; 2019) on the feature data but using the PNN prescriptions as target classes, similar to a binary or multiclass classification task. We call these Mirrored OCTs. Remarkably, the performance of the Mirrored OCTs is equally strong as that of the original PNNs, with a decrease in improvement of only 3.38% on average across the tabular datasets; this implies that interpretability may be recovered with minimal cost to performance.

4. Finally, we demonstrate that our multimodal PNN models achieve stability across randomized data splits comparable to other prescriptive methods and produce realistic prescriptions across the different datasets.

## 2 Methods

In this section, we review the methodology of our PNNs. We first formally define the prescriptive problem we seek to solve (Section 2.1), and then we present the training process, which is divided into four main steps: embedding extraction (Section 2.2), counterfactual estimation (Section 2.3), prescription policy learning (Section 2.4), and interpretability recovery (Section 2.5).

### 2.1 Problem definition

Formally, we consider a prescription problem, which can be characterized by observational data in the form $\{(\boldsymbol{x}_i, y_i, t_i)\}_{i=1}^n$:

- **Features** $\boldsymbol{x}_i \in \mathbb{R}^p$ is the $p$-dimensional feature data for the $i$-th observation.

- **Treatment** $t_i \in \mathcal{T}$ is the treatment applied historically to the $i$-th observation, where $\mathcal{T}$ is the set of all possible treatments. As treatments may be discrete or continuous, there are four possible treatment scenarios: binary (treatment or no treatment), multiple discrete (two or more treatment options), single continuous (one treatment option with continuous values), or multiple continuous (two or more treatment options, some or all taking on continuous values).

- **Outcome** $y_i \in \mathbb{R}$ is the result observed after treatment $t_i \in \mathcal{T}$ has been applied to the $i$-th observation.

Given this observational data, the aim is to develop an optimal prescriptive model that outputs a treatment $t \in \mathcal{T}$ that results in an optimal outcome $y$ for each input observation $\boldsymbol{x}$.

### 2.2 Embedding Extraction

The first step in the model pipeline is to extract embeddings from structured and unstructured data.

### 2.2.1   Structured data

We extract embeddings from structured feature data using traditional pre-processing techniques as described below, where the technique depends on whether the feature is numerical, categorical, or ordinal.

- **Numerical features.** Numerical features are normalized to the interval [0,1] by subtracting the minimum feature value and dividing by the feature range; we do this to increase stability and equal weighting of features during counterfactual estimation and prescriptive modeling. We note that since tree models are independent of data scale, we use the original feature values when training all tree models, which ensure interpretability in the tree splits.

- **Categorical features.** For categorical features, we use one-hot encodings to convert them to binary features, such that each category becomes a new indicator feature.

- **Ordinal features.** Ordinal features are categorical features whose values carry numerical information. Since these categories have a natural order to them, we can assign each category a number such as 1 to 5, where relative magnitude holds information. The feature value assigned to the number "1" conveys that that value is less than that of a feature value assigned the number "4." These ordinal features are then treated as numerical features in our experiments.

### 2.2.2   Unstructured data

We extract embeddings from unstructured data using pretrained, deep learning models. By passing each observation's unstructured data through these pretrained models, we can obtain a vector representation of the unstructured datapoint. In particular, our experiments on medical data in Section 3 use Clinical Longformer (Li et al., 2022), a long sequence transformer model trained via a sparse attention mechanism on domain-specific, large-scale clinical corpora; from this model, we obtain a 768-dimensional embedding vector for each observation's text data.

An important aspect of the multimodal component lies in handling the extracted embeddings. When the dimensionality of these embeddings is high relative to the dataset size, it can lead to overfitting and training instability (Advani et al., 2020). To mitigate this, we explore dimensionality reduction techniques such as Principal Component Analysis (PCA), as well as extracting a compact representation from an intermediate layer of a classification head fine-tuned on the outcome. Though optional, this step can improve training stability, tractability, and downstream performance in our PNN model. In particular, we reduce the clinical note embeddings to 32 dimensions in our medical experiments, though this number can be adjusted depending on the application and dataset size. For the PCA-based reduction, the selected dimensions retain more than 95% of the original variance in all datasets considered.

While we specifically use ClinicalLongformer, any pretrained large language model (LLM) may be used to process unstructured text data. Additionally, any pretrained computer vision (CV) model may be used to process unstructured image data. This results in an embedding extraction step for unstructured data that is not only highly accessible, but also highly flexible.

To get the final multimodal embeddings, we concatenate the individual modalities' embeddings to obtain one large embedding vector.

### 2.3   Counterfactual Estimation

The next step is counterfactual estimation. Because the prescriptive problems' dataset only contains historical observational data, the counterfactuals are unknown, e.g. the hypothetical outcomes $y(\boldsymbol{x}_i, t)$ for $t \neq t_i$, for each observation $\boldsymbol{x}_i$. We therefore perform a counterfactual estimation step (Dudik et al., 2011) that estimates the outcomes for each observation under every treatment. This produces a rewards matrix $\Gamma$, where $\Gamma_{i,t}$ is the estimated outcome of applying treatment $t$ to the $i^{\text{th}}$ observation. The estimation process is slightly different for discrete and continuous treatments.

### 2.3.1 Counterfactual estimation for discrete treatments

We use two methods for counterfactual estimation of discrete treatments. The doubly robust method is, however, preferred for almost all of the experiments in Section 3, since it addresses the treatment assignment bias. The two methods are as follows:

1. **Direct Method.** This method directly learns the outcome function $y_t(\boldsymbol{x})$ by training separate models, one for each treatment $t$. During training, each model uses only the subset of the observations that received treatment $t$. These models can be random forests or boosting methods and output an estimated outcome $\hat{y}_t(\boldsymbol{x})$ for when treatment $t$ is hypothetically applied to observation $\boldsymbol{x}$.

2. **Doubly robust estimation.** Because direct estimation is often prone to treatment assignment bias, the doubly robust estimator attempts to mitigate this bias by re-weighting the estimated direct outcomes with propensity score probabilities. This reweighting is expressed in Equation (1), which calculates the doubly robust reward matrix $\Gamma$:

$$\Gamma_{i,t} = \hat{y}_{i,t} + \mathbb{1}\{t_i = t\}\frac{1}{p_{i,t}}(y_i - \hat{y}_{i,t}), \tag{1}$$

   where $\hat{y}_{i,t} = \hat{y}_t(\boldsymbol{x}_i)$ is the estimated outcome of sample $i$ under treatment $t$, $p_{i,t} = \mathbb{P}[t_i = t]$ is the probability that treatment $t$ is assigned to observation $i$ in real life and $y_i$ is the actual outcome of observation $i$. To reduce the potential instability that arises when we divide with the probability $p_{i,t}$, we clip the ones that are smaller than a certain value (Lee et al., 2011). We generally choose a clipping threshold of 0.01-0.05, depending on the resulting rewards' values.

For binary outcomes, we use classifiers for counterfactual estimation, while for continuous outcomes, we use regressors.

### 2.3.2 Counterfactual estimation of continuous treatments

For continuous treatments, we train a regression model to predict the outcome of the $i^{\text{th}}$ observation using as input the observational data $\boldsymbol{x}_i$ and continuous prescribed treatment doses $T_{i,t}$ for each treatment $t$. Then, by discretizing the continuous dose values and only considering a subset of them as valid treatments, we use the trained model to retrieve the estimated outcome for the $i^{\text{th}}$ observation under all valid treatment schemes. This is analogous to most real-world treatment scenarios; when we handle continuous treatments, we always select a subset of the possible ones, since the real-world treatment options need to be finite.

### 2.4 Prescription policy learning through feedforward neural networks

At its core, the architecture of our Prescriptive Neural Network (PNN) is the classical feedforward neural network trained via backpropagation (Rosenblatt (1958); Rumelhart et al. (1986)). Without loss of generality, we assume our goal is to minimize outcomes in the prescriptive problem. The objective of our prescriptive neural network is to minimize total rewards for the prescriptions $\tau(\boldsymbol{x}_i)$ assigned by the network to each observation $\boldsymbol{x}_i$ in the dataset:

$$\min_{\tau(.)} \sum_{i=1}^{n} \sum_{t \in \mathcal{T}} \mathbb{1}\{\tau(\boldsymbol{x}_i) = t\} \cdot \Gamma_{i,t}, \tag{2}$$

proposed by Amram et al. (2022). Because the indicator function is not differentiable, the backpropagation algorithm cannot handle Equation (2) exactly. We therefore "soften" the objective and leverage an approach analogous to that of multi-classification networks. The PNN assigns treatments probabilistically, such that its output layer consists of $|\mathcal{T}|$ neurons, one for each distinct treatment (as multi-classification networks have an output corresponding to each target class). We denote the output vector of the PNN as $\boldsymbol{z} \in \mathbb{R}^{|\mathcal{T}|}$ and apply a softmax activation function to these output neurons to obtain a probability distribution over the distinct treatments.

We obtain the final prescription of the network by finding the treatment $t$ with the highest probability $\mathbb{P}[\tau(\boldsymbol{x}_i) = t] = \sigma_t(\boldsymbol{z})$. This approach is analogous to a classification network, where the predicted class is the

one with the highest probability among the network's output nodes. The tractable objective for our PNN models is therefore:

$$\min_{\tau(.)} \frac{1}{n} \sum_{i=1}^{n} \sum_{t \in \mathcal{T}} \mathbb{P}[\tau(\boldsymbol{x}_i) = t] \cdot \Gamma_{i,t}. \tag{3}$$

### 2.4.1 Convergence properties

The loss function (3) shares key properties with the cross-entropy loss function commonly used in multiclass classification problems. The cross-entropy loss is defined as:

$$\mathcal{L}(\theta) = -\frac{1}{n} \sum_{i=1}^{n} \sum_{t \in \mathcal{T}} y_{i,t} \log \left( \sigma_t(\boldsymbol{z_i}(\theta)) \right), \tag{4}$$

where $y_{i,t}$ is the true label of sample $i$, and $\sigma_t(\boldsymbol{z})$ is the softmax probability for class $t$. The cross-entropy loss is widely used because it is smooth, has bounded gradients, and is Lipschitz continuous. These properties contribute to the efficient convergence of optimization algorithms like SGD and Adam (Kingma & Ba, 2014; Bottou, 2010).

Similarly, the loss function (3) employed in this work exhibits these desirable properties. Specifically:

- Smoothness: The loss function is smooth because it is a linear combination of the softmax probabilities $\sigma_t(\boldsymbol{z})$, which are themselves smooth functions (Bishop & Nasrabadi, 2006).

- Bounded Gradients: The gradients of the loss function are bounded, since the weights $\Gamma_{it}$ are bounded due to clipping (as described in Section 2.3.1) and the derivative of the softmax function is also bounded (Goodfellow et al., 2016).

- Lipschitz Continuity: The loss function is Lipschitz continuous because the softmax function is Lipschitz continuous, and the weights $\Gamma_{it}$ are bounded (Nesterov, 2013).

The primary difference between our loss function and the cross-entropy loss is that our loss function uses weights $\Gamma_{i,t}$ instead of true labels $y_{i,t}$, and it does not include the logarithm of the probabilities. However, these differences do not fundamentally alter the smoothness, boundedness, or Lipschitz continuity of the loss function. As a result, the convergence behavior of our loss function is similar to that of the cross-entropy loss when training feedforward neural networks for multiclass classification problems (LeCun et al., 2015).

Under the assumption of bounded weights in the network, a property typically observed when training with SGD or Adam (Ghadimi & Lan, 2013; Reddi et al., 2019), the network will converge to critical points of the loss function. This is consistent with the behavior observed in standard neural network training with cross-entropy loss (Zhang et al., 2016). Experimental evidence of the convergence properties is presented in the Appendix Section A.1.

### 2.5 Recovering Interpretability with Optimal Classification Trees

For structured datasets, we are able to recover interpretability through the use of knowledge distillation, in which we fit Optimal Classification Trees (OCTs) (Bertsimas & Dunn, 2017; 2019) on the feature data and prescription outputs of the PNN. We present an example of such a Mirrored OCT in Figure 1 (with other examples available in Appendix A.15). This example comes from the REBOA (resuscitative endovascular balloon occlusion of the aorta) application in Section 3.6. In this real-world problem, we aim to minimize patient mortality by either prescribing (treatment 1) or not prescribing (treatment 0) the REBOA treatment. The tree in the figure is fit on the same observational data used to train its corresponding PNN, while the PNN prescriptions are used as target classes. We observe that this tree is very interpretable, and the features chosen for the splits come from our structured, observational data. If a patient (sample) is assigned to a leaf where the prediction is 0, then they are not prescribed the treatment, and if they are assigned to a leaf where the prediction is 1, the REBOA treatment is recommended.

Figure 1: Example of REBOA Mirrored OCT.

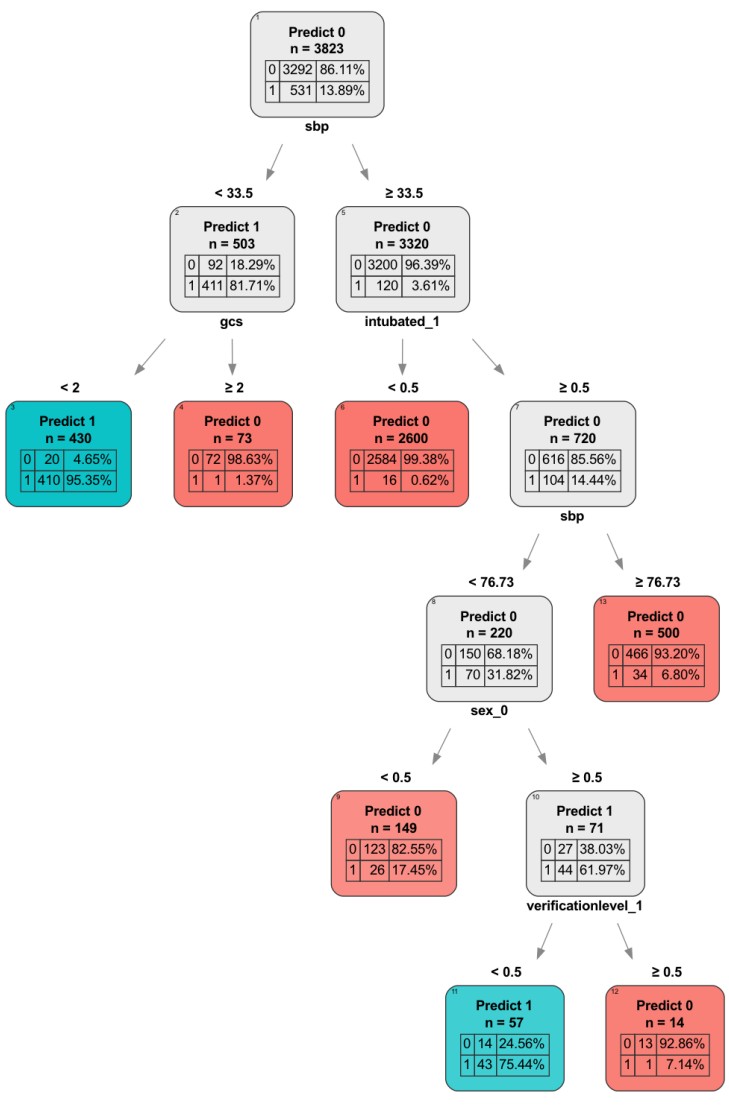

## 3    Experiments with real-world datasets

In this section, we apply PNNs on real-world datasets that are both multimodal and unimodal. We first review methodology for data splits, network architecture, and performance evaluation, which are relevant for all of our experiments. We then report results and relevant discussions for each of our two multimodal datasets and four unimodal datasets.

### 3.1    Data splits

We split each dataset evenly, using 50% for the training and 50% for the test set. This choice is attributed to the fact that in order to evaluate the performance of the prescriptive methods on a test set, knowing the outcomes of the samples under the different treatments is required. Furthermore, to prevent data leakage between training and test splits and ensure fair evaluation, the counterfactual estimation for the test set is necessary and performed separately from that of the training set. This 50-50 split approach follows that

of Amram et al. (2022); Bertsimas et al. (2019) and ensures enough data points in both the training and test sets to to obtain high-quality counterfactual estimates; the typical ratios of 80-20 or 70-30, although possible, could lead to less reliable results on the test set. All results presented in the main text of this work are from the 50-50 train-test split. For completeness, we repeat our experiments with splits of 60-40, 70-30, and 80-20, and we include these results in Appendix Sections A.2.1 and A.2.2 for unstructured and structured datasets, respectively.

### 3.2 Reward estimation

To ensure the robustness of our PNNs across different reward estimation methods and not just the doubly robust method, we train PNNs and Mirrored OCTs with rewards estimated using TARNet (Shalit et al., 2017) and DragonNet (Shi et al., 2019) for the unstructured datasets, and using Causal Forests (Wager & Athey, 2018) for the structured datasets.

All results in the main text are from models trained and reported on rewards estimated using the doubly robust method. We include the results for the other reward estimation methods in Appendix Sections A.3.1 and A.3.2 for the unstructured and structured datasets, respectively, where these results are also reported on the rewards estimated using the doubly robust method for consistency.

### 3.3 Network Architecture

We uniquely tune the PNN architecture for each dataset. We specify and finetune the hyperparameters of the PNN in Appendix A.5. In addition to these architectural choices, we choose the Adam optimizer. We tune the aforementioned hyperparameters using a validation set we extract from the training set that is not used in model training. We typically keep 15% of the training data for the validation set. The Mirrored OCTs are then trained on the prescriptions from each of the PNN models.

### 3.4 Experiments

To evaluate the models' performance on each dataset, we perform multiple train-validation-test splits and report the average performance of each split's models. This ensures that the results are not tailored to a specific data split, and also enables the investigation of stability of the prescriptive methods. Also, given the randomness often associated with training machine learning models, we train multiple models per split and also average their performance. In total, we perform 5 randomized data splits per dataset, and we train 5 models per split, so we train 25 models in total, for each model type.

All methods were implemented in a comparable pipeline, with detailed timing results reported in Appendix Section A.6. In brief, Regress & Compare (XGBoost-based) is consistently the fastest method, while PNNs require longer training time due to gradient-descent optimization; tree-based prescriptive models (OPTs) and Mirrored OCTs fall in between.

### 3.5 Performance metrics

To assess model performance, we use a relative outcome improvement metric measured on the unseen test set. This metric compares the estimated outcome of the treatment prescribed by the model with that of the real-life treatment. For most datasets, where doubly robust estimation is applied, test-set reward matrix entries lack natural meaning. Thus, instead of comparing with the actual outcome and to ensure fairness, hypothetical outcomes for both treatments (model-prescribed and real-life) are drawn from the reward matrix and then compared. The average relative outcome improvement for minimization problems is then computed as:

$$\bar{I} = \frac{\sum_{i=1}^{n}(\Gamma_{i,t_i} - \Gamma_{i,\hat{t}_i})}{\sum_{i=1}^{n}\Gamma_{i,t_i}}, \tag{5}$$

where $\Gamma_{i,\hat{t}_i}$ is the estimated outcome for the $i^{\text{th}}$ observation if the prescribed treatment $\hat{t}_i \in \mathcal{T}$ is applied and $\Gamma_{i,t_i}$ is the estimated outcome for the $i^{\text{th}}$ observation under the real-life treatment $t_i$. For maximization problems, the sign is the opposite.

For the case of unstructured datasets, the evaluation on the test set can be performed using counterfactuals that have been calculated either by using a single modality, or multimodal data. To address that, the outcome improvement is evaluated using the test set counterfactuals with both types of models (in this case, tabular, and multimodal, from tabular and notes).

All evaluations rely on estimated counterfactuals, as real-world outcomes under alternative treatments are unobservable by definition. This is a standard approach in off-policy evaluation for causal inference (Komorowski et al., 2018; Amram et al., 2022; Louizos et al., 2017), and enables fair model comparison under consistent reward estimators. Nonetheless, these estimates reflect model-dependent bias and should not be interpreted as real-world outcome rates. We aim to mitigate this bias as much as possible, by employing independent reward estimators for the test set, and by using multiple types of counterfactual estimators to train the Neural Network, which we report in Appendix A.3.

### 3.6 Unstructured datasets

We demonstrate the efficacy of our PNN models on two real-world, multimodal datasets: transcatheter aortic valve replacement (TAVR) and liver trauma injuries. Both datasets include tabular (structured) and clinical notes (unstructured) data, and we train two models: unimodal models – fit on just the tabular data – and multimodal models – fit on the combined tabular and notes data. More details on how the notes are obtained can be found in the Appendix Section A.13. We report the results of both datasets and discuss the improved performance of the multimodal pipeline in this section.

**Transcatheter aortic valve replacement (TAVR).** Transcatheter aortic valve replacement (TAVR) is a treatment option for patients with severe aortic stenosis across all levels of surgical risk. In the United States, two transcatheter heart valves (THV) are used, the balloon-expandable Edwards Sapien 3 and the self-expanding Medtronic Evolut Pro Plus. Selection of valve choice by medical professionals is generally based on several factors including operator preference, patient characteristics, and valvular/annular anatomy on a computed tomography (CT) scan (Mitsis et al., 2022; Leone et al., 2023). Despite improvement in TAVR devices, implant techniques, and operator experience, permanent pacemaker implantation (PPI) continues to remain a frequent complication with an estimated prevalence of 7-18% (Webb & Wood, 2012; Smith et al., 2011), with potential consequences on patients' mortality and cost of care. This dataset contains demographic (e.g. age, sex, bmi) and medical information (e.g. hypertension, Left Ventricular Ejection Fraction), as well as radiology reports of echocardiograms and CT scans from 2,127 patients, and the problem we seek to solve is prescribe the most appropriate type of valve to patients, so that their risk of PPI is minimized. We train two different sets of models, one where only the tabular features are considered, and one where notes are also incorporated, in the form of embeddings, extracted as described in Section 2.2.2.

**Liver trauma injuries.** Acute liver injury is considered one of the two most common solid organ injuries in blunt trauma victims. However, inaccuracies exist in the grading of liver injuries by human read and interpretation of CT scans, which may lead to mistreatment (Georg et al., 2014). Therefore, personalized treatment for the patient is important in trauma management. This dataset comes from electronic medical records of 722 liver injury patients and includes features such as patient demographics, history of illness, lab results, and allergies. We aim to prescribe either surgical or non-surgical intervention to minimize patient mortality (binary outcome).

In both datasets, to assess the impact of multimodal augmentation and to ensure there is no bias stemming from the modality of reward estimation in the test set, we train separate reward models on the tabular and multimodal datasets. We then evaluate all models on both reward estimations for a fair comparison. In the multimodal case, rewards are always estimated using the full-dimensional embeddings from the pretrained language model. We then train PNNs using three variants of the text embeddings: the full embeddings, a reduced representation from a fine-tuned classification head, and PCA-reduced embeddings. We report performance using the models trained on the type of embeddings that yield the best results for each dataset

on the validation set; for the TAVR dataset, these are the classification head embeddings, and for the liver injury dataset, these are the PCA-reduced embeddings. The final improvements are reported in Table 1.

For both datasets and particularly in the TAVR dataset, we observe that the multimodal models generally outperform the tabular models across both tabular and multimodal reward estimators, demonstrating the benefit of increased information from the added language modality. Although the discrepancy between the outcome improvement can be quite different depending on the modality used to train the test set reward estimators, we observe a pattern of improvement when multimodality is employed under both estimators. The results are also stable, across 5 different data splits and 5 different models per split, and indicate the prescriptive power multimodality can offer.

Mirrored OCTs, trained on the PNNs' prescriptions, result in an outcome improvement comparable or even better to the PNNs, in both datasets, demonstrating that Mirrored OCTs do not generally result in performance decrease. We also quantify the approximation error between the PNN and OCT prediction using the training accuracy of the OCTs on the prescriptions, which we present in Table 2. For example, a training accuracy of 79% means that in the training data split, the OCT correctly classified 79% of the PNN's prescriptions. We observe that the accuracy of the liver dataset's tabular and multimodal Mirrored OCTs is similar, whereas for the TAVR dataset, the multimodal Mirrored OCTs have higher training accuracy than the tabular ones.

Dimensionality reduction is overall beneficial, as full embeddings produce less stable results across all reward estimation methods (see Appendix A.2.1 and A.3.1). We also observe that PNNs and Mirrored OCTs provide improvement in outcomes regardless of the reward estimation method; A.3.1 shows that there are improvements in outcomes of both datasets under TARNet and Dragonnet rewards. More detailed experiments on the uncertainty of the performance estimation for the TAVR and liver trauma datasets can be found in the Appendix Section A.4.1.

Table 1: Improvement (%) in estimated outcome rewards (test set) for the experiments with unstructured data (TAVR, liver trauma), where lower outcome rewards are better. We report the average improvement and standard error across the five splits. Estimator refers to the reward estimation method, using either tabular features only or both tabular and full note embeddings (multimodal).

| Estimator | Method | TAVR models | | Liver trauma models | |
|---|---|---|---|---|---|
| | | Tabular | Multimodal | Tabular | Multimodal |
| Tabular | PNN | $5.05 \pm 2.60$ | $\mathbf{17.87 \pm 6.24}$ | $14.85 \pm 4.39$ | $21.74 \pm 1.96$ |
| | Mirrored OCT | $\mathbf{7.67 \pm 3.45}$ | $17.14 \pm 7.41$ | $\mathbf{26.77 \pm 1.61}$ | $\mathbf{26.46 \pm 1.77}$ |
| Tabular | PNN | $21.09 \pm 1.08$ | $\mathbf{42.89 \pm 5.64}$ | $23.14 \pm 1.66$ | $25.25 \pm 3.00$ |
| & Notes | Mirrored OCT | $\mathbf{22.58 \pm 2.50}$ | $41.66 \pm 7.00$ | $\mathbf{29.14 \pm 2.27}$ | $\mathbf{29.15 \pm 2.17}$ |

Table 2: Training accuracy (%) of the Mirrored OCTs for the unstructured datasets. We report the average accuracy and standard error across the five splits.

| Dataset | Tabular model | Multimodal model |
|---|---|---|
| TAVR | $79.06 \pm 2.23$ | $92.47 \pm 3.14$ |
| Liver trauma | $86.77 \pm 0.78$ | $85.30 \pm 0.85$ |

## 3.7 Structured datasets

We now apply our PNN models to four real-world, unimodal tabular datasets: diabetes management, groceries pricing, splenic injuries treatment, and REBOA in blunt trauma patients. Because these are purely tabular datasets, we are able to recover interpretability by fitting Mirrored OCT models. For more details on the treatment scenarios covered by these four datasets and the counterfactual estimation method employed, please refer to Appendix Table 24.

Table 3: Improvement (%) in estimated outcome rewards (test set) for the experiments with structured datasets. We report the average improvement and standard error across the five 50-50 splits.

| Method | Diabetes | Groceries | Spleen | REBOA |
|---|---|---|---|---|
| Regress & Compare | $2.90 \pm 0.46$ | $94.17 \pm 6.25$ | $8.46 \pm 2.06$ | $-19.69 \pm 16.04$ |
| Causal Forest | $1.60 \pm 0.47$ | $98.68 \pm 5.98$ | $2.43 \pm 4.57$ | $-19.31 \pm 5.16$ |
| Optimal Policy Tree | $2.55 \pm 0.52$ | $106.58 \pm 2.38$ | $12.98 \pm 1.23$ | $17.17 \pm 3.68$ |
| PNN | $\mathbf{3.15 \pm 0.51}$ | $\mathbf{110.88 \pm 1.18}$ | $\mathbf{13.52 \pm 1.74}$ | $17.87 \pm 3.88$ |
| Mirrored OCT | $3.06 \pm 0.53$ | $110.22 \pm 6.94$ | $9.47 \pm 1.91$ | $\mathbf{18.09 \pm 3.18}$ |

Table 4: Training accuracy (%) of the Mirrored OCTs for the structured datasets. We report the average accuracy and standard error across the five splits.

| Diabetes | Groceries | Spleen | REBOA |
|---|---|---|---|
| $86.09 \pm 1.35$ | $98.03 \pm 0.36$ | $92.76 \pm 0.86$ | $95.72 \pm 0.37$ |

We present results for all four structured datasets in Table 3, where we directly compare PNNs and their Mirrored OCTs with the performance of other well-known, state-of-the-art prescriptive methods, including Optimal Policy Trees, Regress & Compare, and Causal Forests. We note that for Regress & Compare, we typically train an XGBoost Regressor or Classifier (depending on the nature of the outcome). For this purpose, we append the actual treatment as a separate column in the observational data and we train the predictive model to predict the real-life outcome under the treatment. To select the best treatment for a new sample, we append each of the available treatments separately and we obtain the final outcome in each case. The treatment that results in the best outcome is selected. We report the training accuracy of the Mirrored OCTs for these structured datasets in Table 4.

**Diabetes management.** This dataset is based on electronic medical records of 58,200 patients with type 2 diabetes from 1999 to 2014 from the Boston Medical Center. It contains information regarding patient demographics, a timeseries of insulin levels, as well as current drug prescriptions. Patient treatments include combinations of insulin, metformin, and oral blood glucose regulation agents, and patient outputs are the resulting hemoglobin A1C measurements (continuous outcome), for which lower values are more optimal.

**Groceries pricing.** For this study, we select the publicly-available retail dataset "The Complete Journey" (Lugauer et al., 2020; Biggs et al., 2021), which contains household-level transactions of many products over two years of 2,500 frequent-shopper households. We focus on one specific product, strawberries. The task here is to, given household demographics, prescribe optimal prices to strawberries with a binary outcome indicating if the household purchases strawberries or not after being assigned the strawberry price. The objective is to maximize revenue, where revenue is defined as the price if strawberries are purchased and zero otherwise. After filtering the data to only the relevant households that had purchased strawberries at least once, the final dataset consists of 97,295 transactions. We impute strawberry prices for cases where strawberry-purchasing households did not purchase strawberries on that specific trip by using the mode of the strawberry prices on the most recent day prior to the trip on which no strawberries were purchased. We consider prices from $2 to $5, inclusive, in increments of $0.50. Since there does not seem to be a strong correlation between strawberry price and the covariate features, rewards are estimated using the direct method.

**Splenic injuries treatment.** The spleen is an immunologic intra-abdominal organ on the left side of the body, which may be removed in the case of injury. In the 1970's to 1980's, the medical community saw a shift towards preservation of the spleen rather than removal, thus making it important to correctly determine if spleen removal was indeed necessary. This specific dataset includes data on spleen surgical operations, in addition to demographic and medical data consisting of numerical, binary, and categorical types. After preprocessing, we have 35,954 rows of patient data in this dataset. We aim to optimally prescribe

splenectomy, angioembolization, or observation in blunt splenic injuries to minimize patient mortality (binary outcome).

**REBOA in blunt trauma patients.** The use of resuscitative endovascular balloon occlusion of the aorta (REBOA) for control of noncompressible torso hemorrhage continues to be highly debated. Being able to appropriately determine if such a treatment should be used is critical in order to decrease the misuse of the treatment in hemodynamically unstable blunt trauma patients. This dataset includes 9,998 patients, with features that are both demographic and medical in nature, including numerical, binary, and categorical values. The goal is to prescribe the REBOA treatment or not to minimize patient mortality (binary outcome). Some feature columns contain unknown values; we therefore use Optimal Imputation (Bertsimas et al., 2017b) with K-Nearest Neighbors to fill the missing values. A few features are integral, and we round imputed values to the nearest integer to maintain integrality.

As shown in Table 3[1], we observe consistent improvements in estimated outcome rewards across all structured datasets and methods. Across datasets, PNNs and Mirrored OCTs are consistently at least one of the two best methods for each dataset. We observe similar results in our additional experiments in Appendix A.2.2, where we also conduct paired significance tests between top-performing and next-best methods, and in Appendix A.4.2, where we quantify the uncertainty in our performance estimation. Using 60-40, 70-30, and 80-20 training-test splits, PNNs and Mirrored OCTs are either best or second best on average, and they outperform other methods with statistical significance in some of the groceries and spleen splits. Similarly, with rewards estimated using causal forest models, PNNs perform comparably to the rest of the methods (see Appendix A.3.2).

The strong and consistent performance of PNNs across datasets highlights their robustness, with results that are statistically comparable or better than other methods. Beyond performance, the key strength of PNNs lies in their flexibility: the same architecture is directly applicable to multimodal datasets. This makes PNNs particularly appealing in settings where both structured and unstructured data must be handled jointly within a unified prescriptive framework.

## 4 Discussion

### 4.1 Relevant Causal Inference Topics

Before estimating counterfactual outcomes, we must ensure that the data satisfy the assumptions required for causal identification. This subsection outlines those assumptions and presents our strategy for enforcing them via diagnostic-informed trimming.

#### 4.1.1 Causal Identification Assumptions and Trimming Strategy

Before estimating causal effects, we adopt two fundamental assumptions to ensure identifiability: the Stable Unit Treatment Value Assumption (SUTVA) and ignorability. SUTVA posits that each unit's outcome is only affected by the treatment assigned to that specific unit, and not by the treatment assignments of other units. Ignorability assumes that all confounding is captured by the observed covariates $X$, formalized as:

$$T \perp\!\!\!\perp \{Y(t)\}_{t \in \mathcal{T}} \mid X,$$

a widely used assumption introduced by Rosenbaum & Rubin (1983). Under this assumption, identification further requires the *positivity* (or overlap) condition:

$$0 < P(T = t \mid X = x) < 1 \quad \forall t \in \mathcal{T},\ x \in \mathrm{supp}(X).$$

Violations occur when the generalized propensity score $e_t(x) = P(T = t \mid X = x)$ approaches 0 or 1, leading to unstable inverse probability weights or model-based extrapolation (Rosenbaum & Rubin, 1983; Petersen et al., 2012).

---

[1]For the groceries dataset, improvement is computed as mean revenue improvement rather than outcome improvement, where mean revenue is $\hat{p}_r = \frac{1}{n} \sum_{i=1}^{n} \Gamma_{i,\hat{t}_i} \cdot \hat{t}_i$, and actual estimated revenue is $p_r = \frac{1}{n} \sum_{i=1}^{n} \Gamma_{i,t_i} \cdot t_i$, with $\hat{t}_i$ the prescribed treatment for sample $i$ by the model, and $t_i$ the real-life treatment.

To mitigate these risks, we perform dataset-specific trimming to discard points with extreme propensity scores before estimating counterfactual outcomes. Our trimming strategy is informed by two diagnostics: Average Overlap (AO) and Propensity Score Distribution.

We compute propensity scores using a Random Forest classifier trained on one of the 50-50 training splits of each dataset. Kernel density estimates (KDEs) of the resulting scores for each treatment group are shown in Appendix Section A.9.

**Average Overlap (AO) Diagnostic.**  To quantify overlap in the multi-treatment setting, we estimate each treatment's propensity score density $f_t(s)$ on $[0, 1]$ using Gaussian KDE. For every treatment pair $(t_i, t_j)$ and target treatment $t$, we define the pairwise overlap:

$$O^{(t)}_{t_i, t_j} = \int_0^1 \min\big[f^{(t)}_{t_i}(s),\, f^{(t)}_{t_j}(s)\big]\, ds.$$

We then compute the overall Average Overlap score:

$$\text{AO} = \frac{1}{|\mathcal{T}|\,\binom{|\mathcal{T}|}{2}} \sum_{t \in \mathcal{T}} \sum_{t_i < t_j} O^{(t)}_{t_i, t_j}.$$

AO values near 1 suggest strong overlap; values closer to 0 indicate lack of support across treatment groups. In our main 50–50 train/test splits, AO scores generally range between 0.25 and 0.6, indicating moderate overlap. The groceries dataset is a notable exception, achieving an AO score near 0.95. Given this strong overlap, we do not trim that dataset. In the case of REBOA, the overlap score is exceptionally low due to extreme imbalance: only a small number of individuals received the treatment, resulting in a sharply peaked KDE and low intersection. Nevertheless, we include REBOA in our analysis because it represents a realistic and high-stakes medical scenario. To mitigate risk of overinterpretation, we interpret REBOA results cautiously.

**Propensity Score-Based Trimming.**  We next examine the extremity of the propensity scores themselves. Based on empirical experimentation and related literature, we adopt fixed thresholds of 0.1 and 0.9 to identify extreme values, since this range has been shown to approximate the optimal rule (Crump et al., 2009). For the unstructured TAVR and liver injury datasets, all scores fall within this range, so we avoid trimming.

For the rest, namely Diabetes, Spleen, and REBOA, we perform dynamic, dataset-specific trimming. Trimming ensures that subsequent causal effect estimation, whether using inverse probability weighting, doubly robust estimators, or model-based methods, is performed within regions of sufficient covariate support. This process reduces the influence of extreme samples and stabilizes estimator behavior while preserving the internal validity of estimated effects in the retained population.

Although the [0.1, 0.9] thresholds are commonly used for trimming (Crump et al., 2009), they are primarily suited for binary treatment settings and may be overly strict in multi-treatment contexts. Prior work (e.g., Stürmer et al. (2021)) shows that percentile-based trimming often achieves better bias–variance tradeoffs than fixed cutoffs, particularly in settings with multiple treatment groups. In our case, we initially evaluated fixed cutoff rules such as [0.1,0.9], but found them overly conservative and difficult to tune across multiple treatment groups, often discarding substantially more data than necessary. Thus, to avoid unnecessary data loss, we instead adopt a percentile-based strategy inspired by Stürmer et al. (2010); Glynn et al. (2019); Stürmer et al. (2021), retaining 90% of each treatment group by removing the lowest and highest 5% of propensity scores (below the 5th percentile or above the 95th percentile). This approach balances enforcing positivity with preserving sufficient data for stable effect estimation.

Post-trimming overlap diagnostics (Appendix Sections A.8, A.10) show that the effect of trimming is not dramatic, especially because our strategy is relatively conservative. This choice aims to minimize dataset alteration. A sufficient number of observations per treatment group is necessary to obtain reliable counterfactual estimates; otherwise, evaluation becomes less trustworthy.

We observe that AO scores are slightly smaller after trimming. This is expected; trimming reduces sample size, which lowers density estimates in kernel smoothing. As a result, the Average Overlap (AO) may decrease even when effective support remains similar. Treatment distributions after trimming are reported in Appendix A.11.

### 4.1.2 Covariate Balance Assessment

To verify that our trimming procedure also improves balance across treatment groups, we assess covariate distributions using the Standardized Mean Difference (SMD) (Rosenbaum & Rubin, 1985). For each covariate $x_j$ and treatment pair $(t, t_0)$, we compute:

$$\text{SMD}_j^{(t,t_0)} = \frac{\left| \mu_j^{(t)} - \mu_j^{(t_0)} \right|}{\sqrt{\frac{1}{2} \left[ \sigma_j^{2\,(t)} + \sigma_j^{2\,(t_0)} \right]}}, \tag{6}$$

where $\mu_j^{(t)}$ and $\sigma_j^{2\,(t)}$ are the sample mean and variance of covariate $x_j$ in treatment group $t$. SMD is a standard diagnostic to assess balance after adjustment (Zhang et al., 2019).

We report average SMDs across all covariates before and after trimming in Appendix Section A.12, for each dataset and treatment pair. We observe that average SMDs remain similar before and after trimming. This is expected, as the proportion of trimmed samples is relatively small in most datasets, and trimming was primarily applied to discard extreme outliers in the propensity score distribution. In particular, although violin plots of the SMD distributions occasionally reveal outliers with higher imbalance, the bulk of the mass remains close to the commonly used 0.1 threshold (Zhang et al., 2019), indicating acceptable covariate balance overall. Trimming thus serves primarily to enforce positivity and support causal identification, rather than substantially altering covariate distributions.

## 4.2 Prescriptions

In general, reporting improvement based on estimated rewards is a good approximation for evaluating the performance of prescriptive methods. However, such metrics do not provide any insights into how realistic the prescriptions are or how different they are from the historical policies. Another critical aspect is model stability. Prescriptive models should be robust in their prescriptions across dataset splits and with respect to inherent randomness during training. Furthermore, for real-world deployment, interpretability is crucial, as users of models must understand where decisions are coming from in order to implement them. We therefore discuss these four topics – realistic and stable prescriptions, interpretability, as well as real-world deployment potential and challenges – in the following sections.

### 4.2.1 Realistic Nature of Prescriptions

Providing realistic prescriptions is crucial, particularly when employing prescriptive tools in practice. To evaluate the realism of the provided prescriptions, we quantify the deviation between the prescribed and real-life treatments of individual samples. This evaluation is carried out per model, by calculating the mean absolute difference between each sample's prescribed and real-life treatment throughout the dataset (training, validation, and test sets) and then averaging it across all samples. For discrete cases (REBOA and spleen datasets), the treatments are ordered in terms of severity, so that the distance is reasonable as a metric. For this purpose, the $N_m = 25$ trained models from each dataset are considered. The mean absolute difference for the $k$-th individual model is given by:

$$D_k = \frac{1}{n} \sum_{i=1}^{n} \|\hat{t}_i - t_i\|_1, \tag{7}$$

where $n$ is the size of the dataset, $\hat{t}_i$ is the prescribed treatment for sample $i$, and $t_i$ is the treatment sample $i$ got in real life. The mean absolute difference across the $N_m = 25$ models is then computed as:

$$\bar{D} = \frac{1}{N_m} \sum_{k=1}^{N_m} D_k. \tag{8}$$

The results are presented in Tables 5 and 6. Clearly, we prefer both high performance $\bar{I}$ and low mean absolute difference $\bar{D}$, as this ensures that improvements in outcomes are not achieved through disproportionate shifts in treatment assignments. We naturally expect that PNN-prescribed treatments are more different than those in real-life, as compared to the other prescriptive methods, since as presented in Sections 3.6 and 3.7, PNNs outperform the other methods in most datasets. Contrary to our expectation, however, we favorably observe that PNNs, as well as the Mirrored OCTs, result in mean absolute difference between the prescribed and the actual treatments that is comparable to the rest of the methods.

Table 5: Mean Absolute Difference between prescribed and actual treatments for structured datasets.

| Method | Diabetes | Groceries | Spleen | REBOA |
|---|---|---|---|---|
| Regress & Compare | **0.4633** | 1.033 | 0.3449 | 0.1770 |
| Causal Forest | 0.7003 | **1.0042** | 0.6294 | 0.1994 |
| Optimal Policy Tree | 0.6966 | 1.0221 | 0.3956 | 0.0496 |
| PNN | 0.5907 | 1.4386 | 0.3261 | 0.0408 |
| Mirrored OCT | 0.5856 | 1.4391 | **0.3193** | **0.0299** |

Table 6: Mean Absolute Difference between prescribed and actual treatments for unstructured datasets. Multimodal here refers to the best version of embeddings for each dataset, namely the classification head embeddings for the TAVR dataset and the PCA-reduced embeddings for the liver injury dataset.

| Method | TAVR models | | Liver trauma models | |
|---|---|---|---|---|
| | Tabular | Multimodal | Tabular | Multimodal |
| PNN | 0.4085 | **0.7841** | 0.3604 | 0.2863 |
| Mirrored OCT | **0.4082** | 0.5616 | **0.3296** | **0.2847** |

For unstructured data, Section 3.6 highlights the advantage of the multimodal approach over tabular-only methods. As expected, this performance edge indicates that the proportion of prescription changes is higher in the multimodal case. The results indicate that, in the TAVR case for example, around 78% of the PNN prescriptions change from one valve to the other in the multimodal case, and 41% in the tabular case, which is a considerable shift. For the liver trauma dataset, the difference between the prescribed and the actual treatments is smaller.

A critical advantage of neural networks, however, is that the user has some control over how much the prescriptions change. For example, depending on the application, a threshold can be selected that limits the number of treatment assignment modifications, and only models that satisfy this constraint on the validation set are considered. Alternatively, one can incorporate a penalty term in the objective function, to penalize an excessive number of treatment switches or to account for different treatment constraints. This is application-specific, but highlights the flexibility that neural networks offer compared to other prescriptive methods.

The realism of prescriptions is also evaluated by examining the average number of distinct prescriptions per type of model, which shows how much the model is capable of utilizing the full treatment space. This is evaluated as:

$$\bar{N} = \frac{1}{|\mathcal{T}|} \cdot \frac{1}{N_m} \sum_{i=1}^{N_m} |t : \exists j : \hat{t}_j = t, \; j = 1, \ldots, n|, \tag{9}$$

where the average number of prescriptions is normalized by the size of treatment space, to calculate a percentage and thus make the metric comparable across the different datasets. The results are presented in Tables 7 and 8.

Table 7: Percentage of different prescriptions selected by the models for structured datasets.

| Method | Diabetes | Groceries | Spleen | REBOA |
|---|---|---|---|---|
| Regress & Compare | 28.33 | 23.33 | 73.33 | 70.0 |
| Causal Forest | **99.67** | **95.33** | **100.0** | **100.0** |
| Optimal Policy Tree | 81.67 | 48.67 | 85.33 | **100.0** |
| PNN | 23.33 | 49.33 | 66.67 | **100.0** |
| Mirrored OCT | 23.33 | 49.33 | 66.67 | **100.0** |

Table 8: Percentage of different prescriptions selected by the models for unstructured datasets. Multimodal here refers to the best version of embeddings for each dataset, namely the classification head embeddings for the TAVR dataset and the PCA-reduced embeddings for the liver injury dataset.

| Method | TAVR models | | Liver trauma models | |
|---|---|---|---|---|
| | Tabular | Multimodal | Tabular | Multimodal |
| PNN | **98.0** | **100.0** | **100.0** | **100.0** |
| Mirrored OCT | 92.0 | 84.0 | 84.0 | 88.0 |

In most of the structured datasets, we observe that PNNs and Mirrored OCTs prescribe a high percentage of the available treatments, with the exception of the diabetes dataset, where the treatment options are multiple. The Regress & Compare approach mostly underutilizes the treatment space, since it prescribes a small percentage of treatments in most cases; this reveals potential treatment assignment bias that may not be mitigated through this approach. Causal Forests seem to make prescriptions that mostly cover the full treatment regime; however their performance is worse than Optimal Policy Trees and PNNs, as discussed in Section 3.7. In the unstructured case, both tabular and multimodal PNN models almost always employ all of the available treatments. We observe that some of the Mirrored OCTs only prescribed one treatment in the TAVR and liver injury dataset, but most of them prescribed both.

Most importantly, PNNs are flexible in this feature too; by using the dropout mechanism (Srivastava et al., 2014) in the last layers of the network, which is often used to prevent overfitting in neural networks, all of the output nodes are forced to be activated during training. As a result, more areas of the network are used, which empirically shows an increase in selected treatments by the model. The flexibility of PNNs is also underlined by the fact that they provide, for each observation, a probability of each treatment, similarly to a classification problem, where neural networks provide a probability of each class. The results presented for PNNs consider the prescription with the highest probability for each sample. However, one can employ a treatment-specific probability threshold to select the final treatment, like in classification problems; for example, this may be done according to some predefined, meaningful, treatment allocation percentage. This provides the user with some control over the resulting treatment distribution.

Overall, PNNs achieve a balance between performance and realism in prescriptions, and they also reasonably cover the treatment space. These are important factors that make them reliable for practitioners and their leverage in different real-life applications.

### 4.2.2 Stability

Given the randomness that is present when training neural networks, their stability compared to other machine learning models is often criticized (Colbrook et al., 2022). The goal of this section is to compare the stability between the different prescriptive approaches by measuring the standard deviation of each

observation's treatment distribution, which results from the $N_m = 25$ different models that have been trained for each dataset. Ideally, the prescriptions should be consistent across the different model runs and data splits; otherwise the method is very sensitive to the training data distribution, which reduces the credibility of the prescriptions.

For each observation, the standard deviation of its prescriptions across the different $N_m$ models is calculated, and we present averages across the different observations in Tables 9[2] and 10.

Table 9: Standard deviation of each sample's prescriptions distribution across $N_m = 25$ models for structured datasets. For Regress & Compare, since we use XGBoost models, there is no randomness in each split, so the 5 models produce the same prescriptions. This explains why in most of the datasets, Regress & Compare has the lowest standard deviation.

| Method | Diabetes | Groceries | Spleen | REBOA |
|---|---|---|---|---|
| Regress & Compare | **0.1007** | **0.0285** | 0.1225 | 0.3069 |
| Causal Forest | 0.6185 | 0.1667 | 0.4761 | 0.3300 |
| Optimal Policy Tree | 0.4951 | 0.7227 | 0.2242 | 0.0426 |
| PNN | 0.2957 | 1.2874 | 0.1268 | 0.0307 |
| Mirrored OCT | 0.2905 | 0.9453 | **0.1174** | **0.0272** |

Table 10: Standard deviation of each sample's prescriptions distribution across $N_m = 25$ models for unstructured datasets. Multimodal here refers to the best version of embeddings for each dataset, namely the classification head embeddings for the TAVR dataset and the PCA-reduced embeddings for the liver injury dataset.

| | TAVR models | | Liver trauma models | |
|---|---|---|---|---|
| Method | Tabular | Multimodal | Tabular | Multimodal |
| PNN | **0.4232** | **0.3803** | 0.3649 | 0.3260 |
| Mirrored OCT | 0.4512 | 0.4900 | **0.2506** | **0.2185** |

We observe that the standard deviation of PNNs' prescriptions is comparable to the other models across all datasets, which indicates that although training neural networks is associated with inherently more randomness (random weight initialization, stochastic gradient descent), they result in relatively consistent prescriptions across different data splits and different runs, similarly to the more deterministic prescriptive methods. Excluding Regress & Compare, a method with deterministic behavior in each seeded split and therefore smaller standard deviation, PNNs and Mirrored OCTs offer considerably more stability in the prescriptions in the Diabetes, Spleen and REBOA datasets compared to Causal Forests and Optimal Policy Trees.

In particular, for the unstructured datasets in Table 10, the standard deviation of Mirrored OCTs is considerably different from the PNNs' in three out of four scenarios. We attribute this to the fact that Mirrored OCTs were likely not able to fully capture the PNNs' complexity and are therefore simpler in their decision-making rules.

### 4.2.3 Interpretability

We discuss now the interpretability of PNNs that is recovered via knowledge distillation of the Mirrored OCTs. In particular, we discuss the unstructured TAVR dataset, for which we can partially recover interpretability, and for the structured diabetes management dataset. Please refer to Appendix A.14 for similar analyses for our other datasets and Appendix A.15 for model visualizations.

---

[2]For multiple continuous treatments (diabetes dataset), to get the standard deviation for each sample, we first calculate the standard deviation for each drug separately and then we average across the three drugs.

**TAVR.** We discuss the multimodal Mirrored OCT from Figure 21 for the TAVR dataset. While the embedding features from the clinical notes are not interpretable, we can still recover some interpretability through the tabular features selected by the Mirrored OCT. We see that the OCT selects one note feature, as well as the Valve-to-Annular Aortic Valve Area ratio (VDAoVA), age, and the difference between the annular area of the patient's native aortic valve and the area of the prosthetic valve being implanted (Area Oversize). We observe that the OCT provides an insight into the prescriptions made by the PNN, even when it is trained on the multimodal data.

**Diabetes management.** We consider and compare the Mirrored OCT from Figure 23 and the Optimal Policy Tree from Figure 24 for diabetes management. Both models were trained with the same data, methods, and parameters as in Section 3.7. For visualization reasons, Figure 23 displays a portion of the tree. From Figure 23, we can see that the features selected by the tree include "HbA1c_mean" (average pre-prescription blood hemoglobin A1C level), age, BMI, and "pastHbA1c1" (past blood hemoglobin A1C level). For example, if a patient has a past blood hemoglobin A1C level of less than 7.25 and is younger than roughly 56 years old, then they would be prescribed with treatment "4" which corresponds to 0 units of insulin, 1 unit of metformin, and 0 units of oral blood glucose regulation agents. The Optimal Policy Tree (Figure 24) selects similar features to the Mirrored OCT, but is considerably larger.

### 4.3 Real-world deployment potential and challenges

Real-world deployment of PNNs and Mirrored OCTs comes with great potential as well as some challenges. Given their flexibility and ease of training, both models show strong promise for practical use. That said, implementing prescriptive models, regardless of the specific architecture, poses several challenges, including ethical considerations around automated decision-making.

In healthcare settings, for instance, building model trust is a key concern. However, we believe the interpretability of Mirrored OCTs, combined with collaborative model development alongside clinicians, offers a promising path toward deployment. Most importantly, we envision PNNs and Mirrored OCTs as tools to support clinical decision-making, rather than replace it. When the model's recommendation aligns with the clinician's decision, it can offer reassurance; when the two disagree, it can prompt a valuable second look. In a real-world application, the final decision would remain with the clinician, informed by both clinical expertise and the model's guidance.

In retail pricing scenarios, such as in the groceries dataset, one main issue is fairness and consumer welfare. In particular, we acknowledge that policies optimized based on historical transaction data may reflect biases that can contribute to potentially harmful price discrimination for consumers. However, similar to the healthcare discussion above, PNNs and Mirrored OCTs are meant to guide decisions, rather than a strict adherence. In a real-world application, policy makers should consider the policies suggested by the data-based models and perform appropriate analyses on fairness, privacy, and legal compliance in order to finalize their pricing policy.

## 5 Conclusions

With its classification-like feedforward neural network architecture, our PNN framework flexibly handles multimodal data, by easily enabling the incorporation of multiple data sources. Furthermore, it is widely applicable for all treatment scenarios, and has the potential of making a great impact in a variety of settings. As demonstrated in our extensive experiments on multiple real-world datasets, the proposed multimodal PNN consistently performs comparably to, or better than, established prescriptive baselines on unimodal tabular data, while requiring only modest computational resources. By extending prescriptive learning frameworks to multimodal settings through a coherent architectural formulation, our approach offers a practical and extensible tool for deploying prescriptive analytics in diverse environments.

Our approach is not only shown to perform strongly quantitatively, but also to provide realistic and stable prescriptions. The discrepancy between the prescribed and real-life treatment distributions is comparable to the other prescriptive methods. The small standard deviation of each sample's assignments from the models indicates that the networks are stable and robust to different data splits. Also, PNNs offer the advantage of

flexibility since the user can adjust the loss function to provide partial control to the prescriptions, leveraging expert knowledge.

Deep learning methods generally sacrifice interpretability. On unimodal tabular datasets, we are able to recover interpretability through a knowledge distillation approach leveraging interpretable OCT models, and on multimodal datasets, some interpretability may still be recovered. These Mirrored OCTs demonstrate similarly high performance in our real-world experiments, demonstrating that we can maintain high performance without sacrificing interpretability. This recovery of interpretability is critical for real-world deployment of deep learning models.

We conclude that our multimodal deep learning framework, PNNs, offers both flexibility and strong performance, effectively utilizing deep learning to process multimodal data. By integrating multiple data sources, the framework greatly enhances decision-making capabilities. This unified approach demonstrates its potential as a versatile prescriptive tool, well-suited for a wide range of applications.

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

# A Appendix

## A.1 Convergence analysis

To examine the convergence properties of our custom loss function, we track the training and validation loss of PNNs, trained on all the datasets, for different values of the temperature parameter $\tau$ and the entropy regularization weight $\lambda$. More specifically, we consider the general loss function:

$$\mathcal{L}_{\tau,\lambda}(\theta) \;=\; \frac{1}{n}\sum_{i=1}^{n}\sum_{t\in\mathcal{T}}\sigma_t(\boldsymbol{z}_i(\theta),\tau)\Gamma_{i,t} \;+\; \lambda\frac{1}{n}\sum_{i=1}^{n}H(\sigma_t(\boldsymbol{z}_i(\theta),\tau)),\tag{10}$$

where $\boldsymbol{\sigma}(\boldsymbol{z}_i(\theta),\tau) = \mathrm{softmax}\!\left(\frac{\boldsymbol{z}_i(\theta)}{\tau}\right)$ and $H(\boldsymbol{\sigma}(\boldsymbol{z}_i(\theta),\tau)) \;=\; -\sum_{t\in\mathcal{T}}\sigma_t(\boldsymbol{z}_i(\theta),\tau)\log\sigma_t(\boldsymbol{z}_i(\theta),\tau)$.

We conduct two sets of experiments: (1) fixing $\lambda = 0$ and varying the temperature $\tau$ and (2) fixing $\tau = 1.0$ and varying the regularization weight $\lambda$. All experiments are performed on one of the 50/50 train–test splits. For each configuration, we train five models with different random seeds to account for variability in weight initialization and the stochasticity of gradient descent. The averaged loss curves are presented in Figures 2 - 4 and 5 - 7 respectively.

Figure 2: Training and validation loss curves under different temperature values $\tau$ (Diabetes, Groceries).

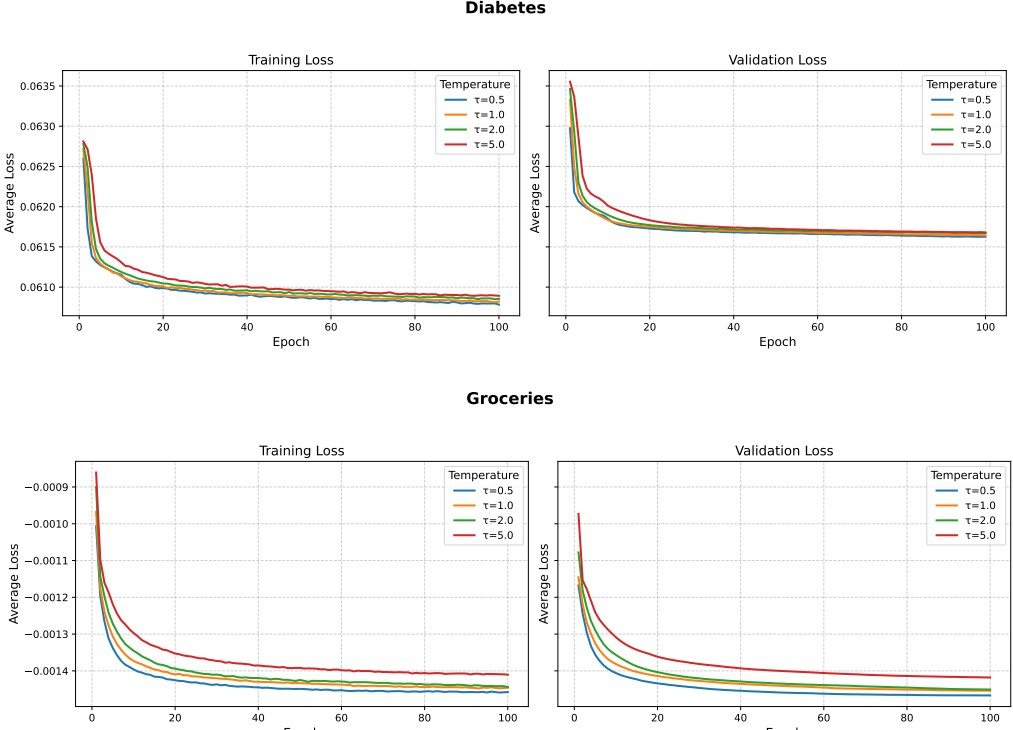

Across all datasets, we observe smooth convergence behavior for both training and validation losses. The TAVR and Liver datasets display slightly more variability, which is expected given their smaller sample sizes and higher feature dimensionality. Convergence is robust for all values of $\tau$, with larger temperatures leading to slower but more stable convergence. Varying $\lambda$ has minimal effect on loss magnitude, except in the Groceries and Spleen datasets, where stronger entropy regularization ($\lambda$) slightly increases the converged loss value.

## A.2 Effect of different split ratios

### A.2.1 Unstructured Data

We present full results for the unstructured datasets under different training and test split ratios (50/50, 60/40, 70/30, 80/20) in Table 11 and the respective Mirrored OCT training accuracies in Table 12. We find that the relative order of the model performance remains similar for the different types of embeddings as in the 50/50 case. Results on smaller test set sizes should be interpreted cautiously, since they are reported on rewards that are trained on a smaller set compared to the 50/50 or 60/40 splits, for example.

Figure 3: Training and validation loss curves under different temperature values $\tau$ (Spleen, REBOA).

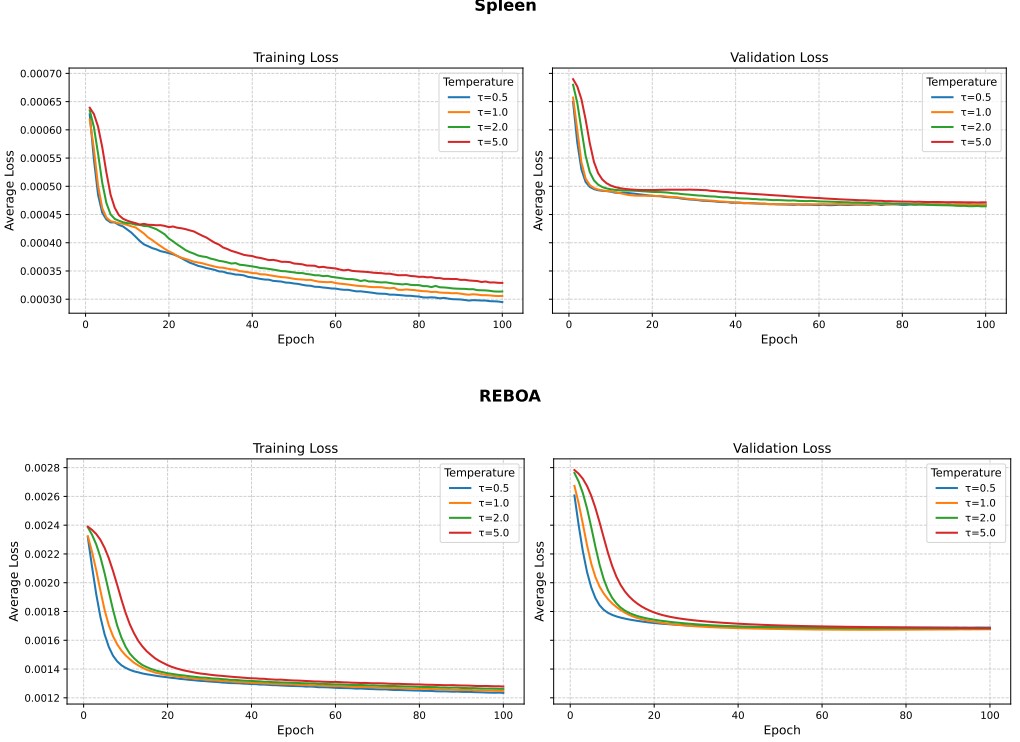

Figure 4: Training and validation loss curves under different temperature values $\tau$ (TAVR, Liver).

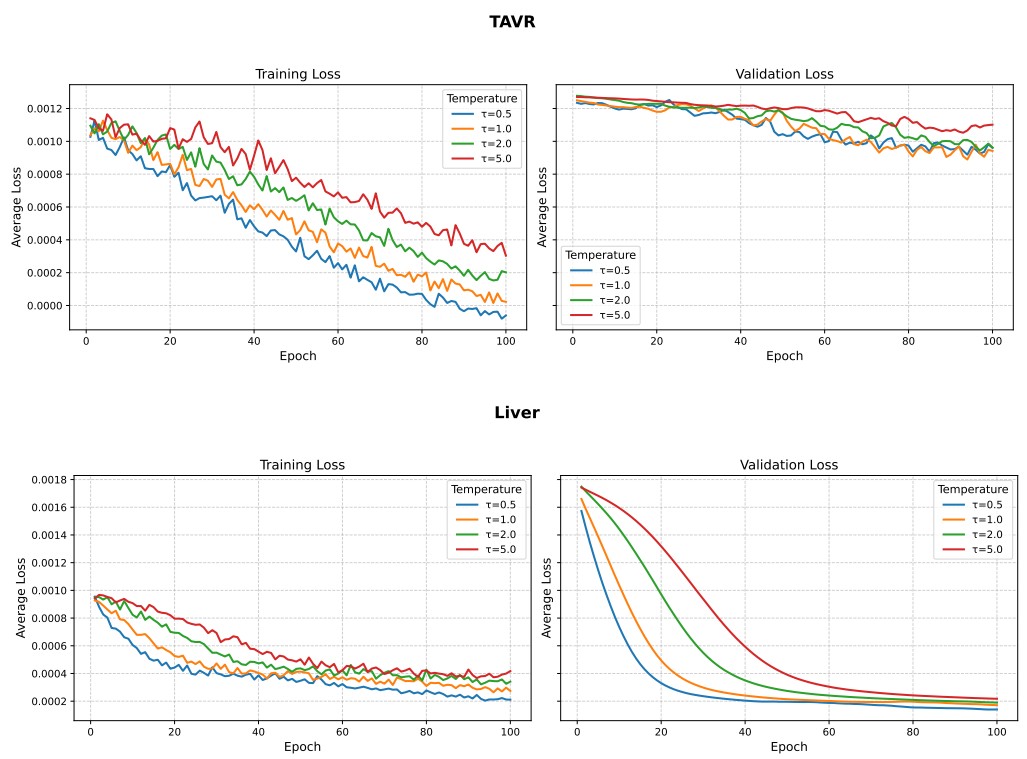

Figure 5: Training and validation loss curves under different entropy regularization weights $\lambda$ (Diabetes, Groceries).

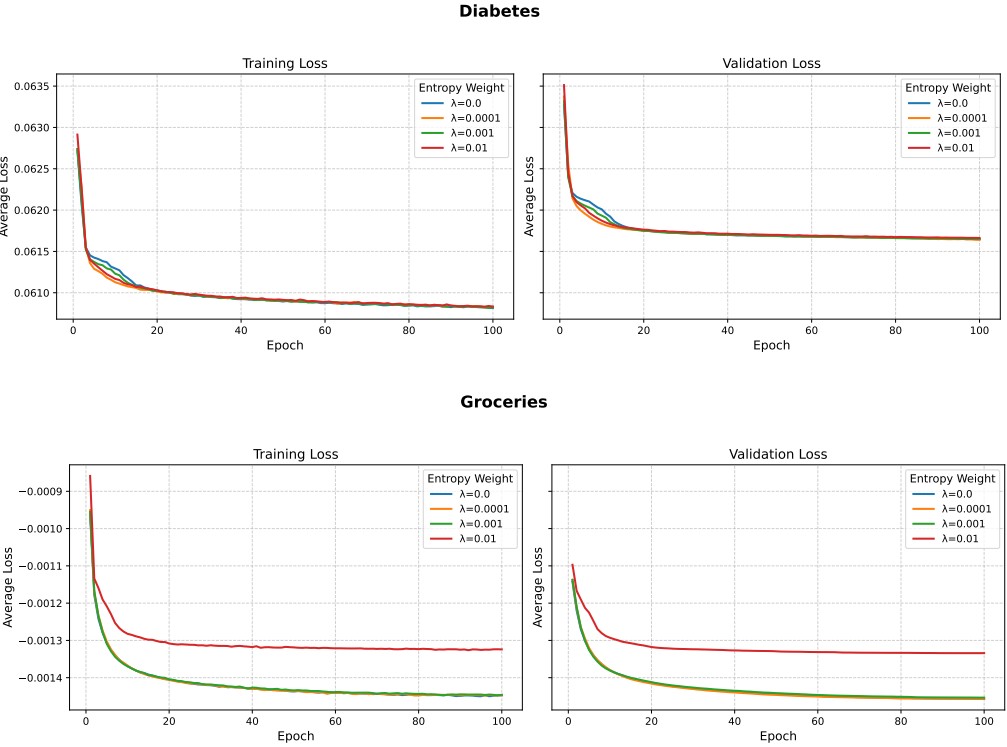

Figure 6: Training and validation loss curves under different entropy regularization weights $\lambda$ (Spleen, REBOA).

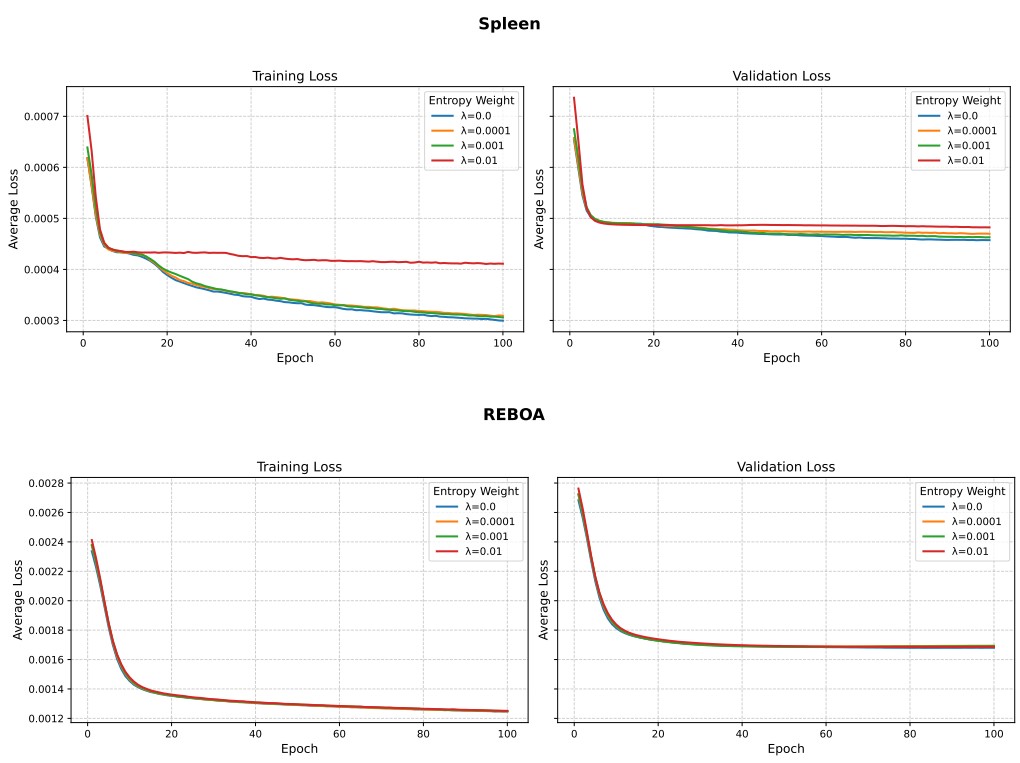

Figure 7: Training and validation loss curves under different entropy regularization weights $\lambda$ (TAVR, Liver).

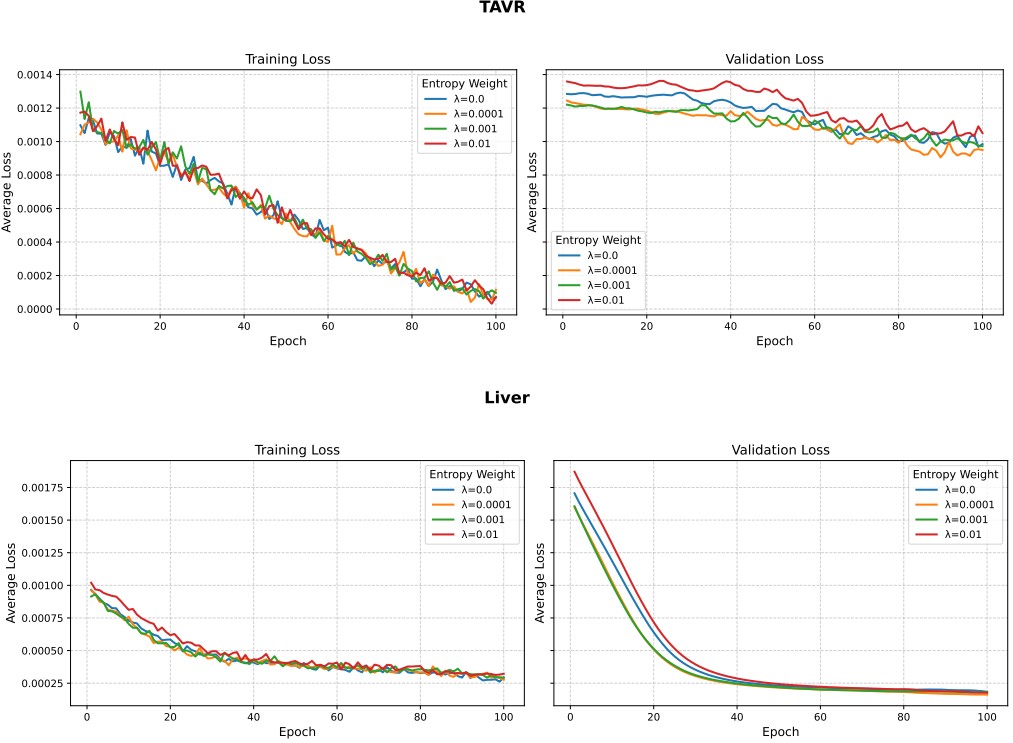

Table 12: Training accuracy (%) of the Mirrored OCTs for the unstructured datasets under different train-test split ratios. We report the average accuracy and standard error across five random train-test splits per split ratio.

| Split Ratio | Data | TAVR models | Liver trauma models |
|---|---|---|---|
| 50/50 | Tab | $79.06 \pm 2.23$ | $86.77 \pm 0.78$ |
| | Full | $81.68 \pm 3.03$ | $90.25 \pm 1.2$ |
| | CH | $92.47 \pm 3.14$ | $88.37 \pm 1.31$ |
| | PCA | $73.19 \pm 0.66$ | $85.30 \pm 0.85$ |
| 60/40 | Tab | $71.07 \pm 1.26$ | $92.09 \pm 1.51$ |
| | Full | $69.16 \pm 0.9$ | $88.3 \pm 2.27$ |
| | CH | $72.3 \pm 1.38$ | $87.98 \pm 1.19$ |
| | PCA | $70.13 \pm 0.61$ | $83.70 \pm 1.57$ |
| 70/30 | Tab | $71.03 \pm 0.74$ | $90.56 \pm 1.38$ |
| | Full | $70.57 \pm 1.08$ | $89.21 \pm 0.84$ |
| | CH | $73.53 \pm 1.29$ | $88.2 \pm 0.77$ |
| | PCA | $70.31 \pm 0.87$ | $82.54 \pm 1.84$ |
| 80/20 | Tab | $72.23 \pm 0.91$ | $93.35 \pm 1.29$ |
| | Full | $70.73 \pm 0.43$ | $88.96 \pm 1.41$ |
| | CH | $77.10 \pm 2.03$ | $87.73 \pm 1.57$ |
| | PCA | $70.01 \pm 1.07$ | $83.82 \pm 2.72$ |

Table 11: Improvement (%) in estimated outcome rewards (test set) for the unstructured datasets under different train-test split ratios. We report the average accuracy and standard error across five random train-test splits per split ratio. Full corresponds to the full embeddings as extracted from Clinical Longformer, CH corresponds to the representation from the Classification Head, and PCA to the embeddings after PCA reduction. Estimator refers to the reward estimation method, using either tabular features only or both tabular and full note embeddings (multimodal).

| Split Ratio | Estimator | Data | TAVR models | | Liver trauma models | |
| | | | PNN | Mirrored OCT | PNN | Mirrored OCT |
|---|---|---|---|---|---|---|
| 50/50 | Tabular | Tab | $5.05 \pm 2.60$ | $7.67 \pm 3.45$ | $14.85 \pm 4.39$ | $26.77 \pm 1.61$ |
| | | Full | $8.59 \pm 3.05$ | $4.26 \pm 4.15$ | $19.10 \pm 6.47$ | $\mathbf{26.79 \pm 3.74}$ |
| | | CH | $\mathbf{17.87 \pm 6.24}$ | $\mathbf{17.14 \pm 7.41}$ | $13.84 \pm 5.35$ | $23.16 \pm 8.44$ |
| | | PCA | $8.53 \pm 3.49$ | $9.69 \pm 5.13$ | $\mathbf{21.74 \pm 1.96}$ | $26.46 \pm 1.77$ |
| | Tabular & Notes | Tab | $21.09 \pm 1.08$ | $22.58 \pm 2.50$ | $23.14 \pm 1.66$ | $29.14 \pm 2.27$ |
| | | Full | $26.19 \pm 2.41$ | $21.46 \pm 5.55$ | $19.65 \pm 2.28$ | $23.37 \pm 4.75$ |
| | | CH | $\mathbf{42.89 \pm 5.64}$ | $\mathbf{41.66 \pm 7.00}$ | $23.94 \pm 3.08$ | $26.16 \pm 4.56$ |
| | | PCA | $26.32 \pm 2.57$ | $26.24 \pm 5.09$ | $\mathbf{25.25 \pm 3.00}$ | $\mathbf{29.15 \pm 2.17}$ |
| 60/40 | Tabular | Tab | $12.86 \pm 2.03$ | $8.00 \pm 1.93$ | $\mathbf{41.26 \pm 8.09}$ | $44.74 \pm 5.78$ |
| | | Full | $12.10 \pm 2.85$ | $8.90 \pm 1.44$ | $23.32 \pm 4.69$ | $26.34 \pm 7.01$ |
| | | CH | $\mathbf{14.67 \pm 2.65}$ | $\mathbf{11.39 \pm 1.51}$ | $35.21 \pm 7.56$ | $39.85 \pm 5.54$ |
| | | PCA | $9.03 \pm 1.48$ | $6.00 \pm 2.12$ | $34.80 \pm 8.76$ | $\mathbf{47.06 \pm 7.63}$ |
| | Tabular & Notes | Tab | $14.69 \pm 2.67$ | $7.52 \pm 1.51$ | $\mathbf{35.94 \pm 4.93}$ | $36.25 \pm 7.05$ |
| | | Full | $12.40 \pm 2.33$ | $9.55 \pm 1.01$ | $14.20 \pm 6.46$ | $25.62 \pm 7.87$ |
| | | CH | $\mathbf{15.71 \pm 3.54}$ | $\mathbf{12.85 \pm 0.87}$ | $25.25 \pm 7.49$ | $26.73 \pm 10.28$ |
| | | PCA | $9.52 \pm 1.82$ | $6.12 \pm 1.66$ | $24.55 \pm 4.35$ | $\mathbf{38.83 \pm 5.54}$ |
| 70/30 | Tabular | Tab | $6.34 \pm 2.54$ | $1.84 \pm 1.39$ | $36.20 \pm 8.16$ | $41.53 \pm 8.65$ |
| | | Full | $\mathbf{8.97 \pm 2.10}$ | $2.33 \pm 3.72$ | $23.97 \pm 5.93$ | $28.35 \pm 4.17$ |
| | | CH | $6.27 \pm 2.31$ | $\mathbf{2.72 \pm 4.25}$ | $\mathbf{39.67 \pm 6.06}$ | $34.63 \pm 5.50$ |
| | | PCA | $8.13 \pm 1.47$ | $1.08 \pm 3.27$ | $33.39 \pm 7.79$ | $\mathbf{43.03 \pm 9.30}$ |
| | Tabular & Notes | Tab | $7.93 \pm 3.02$ | $1.98 \pm 1.98$ | $\mathbf{27.76 \pm 7.34}$ | $33.57 \pm 6.35$ |
| | | Full | $8.51 \pm 2.71$ | $3.34 \pm 2.80$ | $18.18 \pm 6.77$ | $21.80 \pm 3.96$ |
| | | CH | $8.27 \pm 2.28$ | $\mathbf{4.11 \pm 4.13}$ | $23.95 \pm 5.10$ | $25.52 \pm 5.41$ |
| | | PCA | $\mathbf{9.73 \pm 1.72}$ | $1.72 \pm 2.71$ | $23.31 \pm 5.80$ | $\mathbf{34.59 \pm 5.63}$ |
| 80/20 | Tabular | Tab | $\mathbf{9.85 \pm 2.46}$ | $2.34 \pm 2.17$ | $44.93 \pm 3.42$ | $48.28 \pm 3.48$ |
| | | Full | $7.17 \pm 1.15$ | $1.55 \pm 2.98$ | $27.18 \pm 4.39$ | $42.38 \pm 3.90$ |
| | | CH | $8.55 \pm 3.68$ | $\mathbf{5.73 \pm 2.91}$ | $\mathbf{45.21 \pm 2.82}$ | $44.03 \pm 4.32$ |
| | | PCA | $9.61 \pm 3.51$ | $4.03 \pm 3.35$ | $41.95 \pm 4.76$ | $\mathbf{50.89 \pm 6.24}$ |
| | Tabular & Notes | Tab | $\mathbf{9.38 \pm 2.64}$ | $1.67 \pm 2.23$ | $36.22 \pm 3.50$ | $40.88 \pm 4.02$ |
| | | Full | $6.82 \pm 1.12$ | $0.93 \pm 2.84$ | $25.71 \pm 5.57$ | $33.84 \pm 6.52$ |
| | | CH | $7.77 \pm 2.88$ | $\mathbf{6.01 \pm 3.05}$ | $\mathbf{36.96 \pm 2.98}$ | $35.64 \pm 3.52$ |
| | | PCA | $9.34 \pm 3.70$ | $3.27 \pm 2.79$ | $36.42 \pm 4.82$ | $\mathbf{45.13 \pm 6.53}$ |

### A.2.2 Structured data

We present full results for the structured datasets under different training and test split ratios (50/50, 60/40, 70/30, 80/20) in Table 13, the corresponding statistical significance analysis in Table 14 and the Mirrored OCT training accuracies in Table 15. Although the exact improvement varies between the splits, we observe that the relative order of the model performance remains similar, and that PNNs and Mirrored OCTs either outperform or perform similarly to existing models. The paired significance tests confirm that,

in most cases, our proposed models perform comparably to the next-best method, with a few instances showing statistically significant superiority according to paired $t$-tests (denoted $^\dagger$ in Table 14). Because each comparison is conducted over only five random splits, both the paired $t$-test and the permutation test are conservative. Overall, we observe that improvements are modest but consistent; results on smaller test sets should be interpreted with appropriate caution, due to the smaller size of the test set.

Table 13: Improvement (%) in estimated outcome rewards (test set) for structured datasets under different train-test split ratios. We report the average accuracy and standard error across five random train-test splits per split ratio.

| Split Ratio | Method | Diabetes | Groceries | Spleen | REBOA |
|---|---|---|---|---|---|
| 50/50 | Regress & Compare | $2.90 \pm 0.46$ | $94.17 \pm 6.25$ | $8.46 \pm 2.06$ | $-19.69 \pm 16.04$ |
| | Causal Forest | $1.60 \pm 0.47$ | $98.68 \pm 5.98$ | $2.43 \pm 4.57$ | $-19.31 \pm 5.16$ |
| | Optimal Policy Tree | $2.55 \pm 0.52$ | $106.58 \pm 2.38$ | $12.98 \pm 1.23$ | $17.17 \pm 3.68$ |
| | PNN | $\mathbf{3.15 \pm 0.51}$ | $\mathbf{110.88 \pm 1.18}$ | $\mathbf{13.52 \pm 1.74}$ | $17.88 \pm 3.88$ |
| | Mirrored OCT | $3.06 \pm 0.53$ | $110.22 \pm 6.94$ | $9.47 \pm 1.91$ | $\mathbf{18.09 \pm 3.18}$ |
| 60/40 | Regress & Compare | $1.47 \pm 1.76$ | $98.34 \pm 2.70$ | $4.60 \pm 2.32$ | $-3.13 \pm 3.40$ |
| | Causal Forest | $-0.35 \pm 1.88$ | $103.50 \pm 2.71$ | $3.05 \pm 2.88$ | $-15.68 \pm 5.77$ |
| | Optimal Policy Tree | $1.13 \pm 1.55$ | $107.81 \pm 3.45$ | $10.55 \pm 1.24$ | $16.56 \pm 1.90$ |
| | PNN | $\mathbf{1.61 \pm 1.51}$ | $\mathbf{112.67 \pm 3.71}$ | $\mathbf{14.27 \pm 2.24}$ | $16.23 \pm 2.10$ |
| | Mirrored OCT | $1.50 \pm 1.54$ | $111.97 \pm 4.46$ | $9.38 \pm 1.74$ | $\mathbf{17.36 \pm 1.70}$ |
| 70/30 | Regress & Compare | $\mathbf{2.44 \pm 1.06}$ | $100.27 \pm 3.86$ | $2.98 \pm 1.57$ | $4.34 \pm 2.51$ |
| | Causal Forest | $0.54 \pm 0.86$ | $103.16 \pm 1.25$ | $-5.0 \pm 5.61$ | $0.99 \pm 4.89$ |
| | Optimal Policy Tree | $1.36 \pm 2.32$ | $108.47 \pm 1.65$ | $7.47 \pm 2.05$ | $\mathbf{25.49 \pm 2.89}$ |
| | PNN | $1.96 \pm 0.67$ | $\mathbf{114.14 \pm 1.91}$ | $\mathbf{11.31 \pm 1.81}$ | $25.38 \pm 2.88$ |
| | Mirrored OCT | $2.20 \pm 0.79$ | $113.81 \pm 1.90$ | $8.66 \pm 6.46$ | $24.31 \pm 2.66$ |
| 80/20 | Regress & Compare | $\mathbf{1.89 \pm 1.40}$ | $109.72 \pm 2.18$ | $4.89 \pm 2.87$ | $0.50 \pm 3.37$ |
| | Causal Forest | $0.06 \pm 1.39$ | $111.95 \pm 0.85$ | $11.02 \pm 5.48$ | $-5.32 \pm 9.28$ |
| | Optimal Policy Tree | $0.80 \pm 1.36$ | $107.40 \pm 4.45$ | $15.98 \pm 2.16$ | $\mathbf{27.80 \pm 6.27}$ |
| | PNN | $1.82 \pm 1.41$ | $\mathbf{112.30 \pm 7.63}$ | $\mathbf{18.01 \pm 1.34}$ | $27.28 \pm 6.07$ |
| | Mirrored OCT | $1.75 \pm 1.39$ | $110.40 \pm 6.55$ | $10.11 \pm 4.49$ | $26.69 \pm 5.75$ |

Table 15: Training accuracy (%) of the Mirrored OCTs for the structured datasets under different train-test split ratios. We report the average accuracy and standard error across five random train-test splits per split ratio.

| Split Ratio | Diabetes | Groceries | Spleen | REBOA |
|---|---|---|---|---|
| 50/50 | $86.09 \pm 1.35$ | $98.03 \pm 0.36$ | $92.76 \pm 0.86$ | $95.72 \pm 0.37$ |
| 60/40 | $88.1 \pm 0.90$ | $88.82 \pm 1.07$ | $93.13 \pm 0.64$ | $96.77 \pm 0.18$ |
| 70/30 | $86.9 \pm 0.90$ | $90.56 \pm 0.53$ | $91.78 \pm 0.40$ | $96.78 \pm 0.25$ |
| 80/20 | $87.83 \pm 1.42$ | $89.37 \pm 0.51$ | $91.27 \pm 0.55$ | $96.97 \pm 0.40$ |

## A.3 Effect of different types of rewards

### A.3.1 Unstructured Data

We present full results for the unstructured datasets under different reward estimation methods (doubly robust, Dragonnet, and TARNet) for training the PNN in Table 16 and the respective Mirrored OCT training accuracies in Table 17. For consistency, improvements are reported on test set rewards estimated using the doubly robust method. We observe that the magnitude of improvements is similar across rewards,

Table 14: Paired significance tests between top-performing and next-best methods across splits and datasets. Reported $p$-values correspond to paired $t$-tests ($p_t$) and permutation tests ($p_{\text{perm}}$) computed over 5 random seeds per split.

| Split | Dataset | Top-performing method | Next-best method | $p_t$ | $p_{\text{perm}}$ |
|---|---|---|---|---|---|
| 50/50 | Diabetes | PNN | Regress & Compare | 0.385 | 0.438 |
| | Groceries | PNN | Optimal Policy Tree | $0.006^\dagger$ | 0.063 |
| | Spleen | PNN | Optimal Policy Tree | 0.564 | 0.563 |
| | REBOA | Mirrored OCT | Optimal Policy Tree | 0.295 | 0.375 |
| 60/40 | Diabetes | PNN | Regress & Compare | 0.690 | 0.688 |
| | Groceries | PNN | Optimal Policy Tree | $0.006^\dagger$ | 0.063 |
| | Spleen | PNN | Optimal Policy Tree | 0.091 | 0.063 |
| | REBOA | Mirrored OCT | Optimal Policy Tree | 0.072 | 0.125 |
| 70/30 | Diabetes | Regress & Compare | Mirrored OCT | 0.659 | 0.563 |
| | Groceries | PNN | Optimal Policy Tree | 0.071 | 0.125 |
| | Spleen | PNN | Optimal Policy Tree | $0.004^\dagger$ | 0.063 |
| | REBOA | Optimal Policy Tree | PNN | 0.873 | 1.0 |
| 80/20 | Diabetes | Regress & Compare | PNN | 0.384 | 0.500 |
| | Groceries | PNN | Causal Forest | 0.967 | 1.0 |
| | Spleen | PNN | Optimal Policy Tree | 0.253 | 0.250 |
| | REBOA | Optimal Policy Tree | PNN | 0.716 | 0.750 |

increasing the confidence for the quality of the resulting prescriptions. In the case of TAVR, the classification head embeddings seem to result in the best performance, suggesting that supervised fine-tuning can produce more informative and stable embeddings, whereas in the liver case full embeddings seem to perform better under Dragonnet and TARNet rewards, followed by embeddings after PCA.

Table 16: Improvement (%) in estimated outcome rewards (test set) for the experiments with unstructured data under different reward estimation models. We report the average improvement and standard error across five random 50-50 training-test splits. The improvement and error is reported in terms of the doubly robust rewards for consistency. In the Data column, Full corresponds to the full embeddings as extracted from Clinical Longformer, CH corresponds to the Representation from the Classification Head, and PCA to the embeddings after PCA reduction. Estimator refers to the reward estimation method, using either tabular features only or both tabular and full note embeddings (multimodal). We have bolded the greatest improvement for each combination of dataset, model type, and estimator.

| Rewards | Estimator | Data | TAVR models | | Liver trauma models | |
| | | | PNN | Mirrored OCT | PNN | Mirrored OCT |
|---|---|---|---|---|---|---|
| Doubly robust | Tabular | Tab | $5.05 \pm 2.60$ | $7.67 \pm 3.45$ | $14.85 \pm 4.39$ | $26.77 \pm 1.61$ |
| | | Full | $8.59 \pm 3.05$ | $4.26 \pm 4.15$ | $19.10 \pm 6.47$ | $\mathbf{26.79 \pm 3.74}$ |
| | | CH | $\mathbf{17.87 \pm 6.24}$ | $17.14 \pm 7.41$ | $13.84 \pm 5.35$ | $23.16 \pm 8.44$ |
| | | PCA | $8.53 \pm 3.49$ | $\mathbf{9.69 \pm 5.13}$ | $\mathbf{21.74 \pm 1.96}$ | $26.46 \pm 1.77$ |
| | Tabular & Notes | Tab | $21.09 \pm 1.08$ | $22.58 \pm 2.50$ | $23.14 \pm 1.66$ | $29.14 \pm 2.27$ |
| | | Full | $26.19 \pm 2.41$ | $21.46 \pm 5.55$ | $19.65 \pm 2.28$ | $23.37 \pm 4.75$ |
| | | CH | $\mathbf{42.89 \pm 5.64}$ | $\mathbf{41.66 \pm 7.00}$ | $23.94 \pm 3.08$ | $26.16 \pm 4.56$ |
| | | PCA | $26.32 \pm 2.57$ | $26.24 \pm 5.09$ | $\mathbf{25.25 \pm 3.00}$ | $\mathbf{29.15 \pm 2.17}$ |
| Dragonnet | Tabular | Tab | $16.38 \pm 3.83$ | $17.1 \pm 3.59$ | $30.72 \pm 4.93$ | $20.08 \pm 5.53$ |
| | | Full | $19.01 \pm 3.19$ | $20.15 \pm 2.7$ | $\mathbf{33.09 \pm 5.43}$ | $\mathbf{25.51 \pm 5.78}$ |
| | | CH | $\mathbf{22.13 \pm 5.23}$ | $\mathbf{23.16 \pm 3.92}$ | $25.18 \pm 9.90$ | $1.67 \pm 2.0$ |
| | | PCA | $18.71 \pm 2.98$ | $19.69 \pm 4.67$ | $27.79 \pm 6.70$ | $15.21 \pm 7.41$ |
| | Tabular & Notes | Tab | $24.09 \pm 4.77$ | $23.82 \pm 3.86$ | $9.81 \pm 5.19$ | $17.53 \pm 7.07$ |
| | | Full | $27.14 \pm 3.60$ | $27.81 \pm 3.89$ | $\mathbf{29.33 \pm 7.60}$ | $\mathbf{26.35 \pm 5.98}$ |
| | | CH | $\mathbf{36.32 \pm 8.40}$ | $\mathbf{38.48 \pm 7.21}$ | $11.98 \pm 4.13$ | $14.78 \pm 4.6$ |
| | | PCA | $26.67 \pm 2.38$ | $27.21 \pm 4.7$ | $16.24 \pm 6.58$ | $15.8 \pm 8.25$ |
| TARNet | Tabular | Tab | $16.38 \pm 3.93$ | $\mathbf{20.05 \pm 2.76}$ | $29.71 \pm 6.98$ | $35.61 \pm 5.13$ |
| | | Full | $\mathbf{19.29 \pm 1.98}$ | $9.12 \pm 3.59$ | $\mathbf{37.44 \pm 4.72}$ | $34.54 \pm 6.27$ |
| | | CH | $16.97 \pm 5.14$ | $17.19 \pm 4.76$ | $26.50 \pm 11.81$ | $21.23 \pm 11.43$ |
| | | PCA | $16.77 \pm 2.63$ | $17.66 \pm 4.22$ | $25.90 \pm 8.41$ | $\mathbf{38.54 \pm 4.26}$ |
| | Tabular & Notes | Tab | $23.70 \pm 4.94$ | $27.57 \pm 3.24$ | $14.30 \pm 7.27$ | $18.92 \pm 6.35$ |
| | | Full | $24.56 \pm 1.65$ | $24.82 \pm 4.27$ | $\mathbf{30.37 \pm 7.40}$ | $26.35 \pm 5.98$ |
| | | CH | $\mathbf{34.08 \pm 6.90}$ | $\mathbf{34.41 \pm 6.88}$ | $13.76 \pm 6.23$ | $9.19 \pm 6.5$ |
| | | PCA | $23.97 \pm 2.86$ | $25.01 \pm 4.32$ | $15.60 \pm 7.08$ | $\mathbf{25.0 \pm 6.6}$ |

Table 17: Training accuracy (%) of the Mirrored OCTs for the unstructured datasets under different reward estimation models. We report the average accuracy and standard error across five random 50-50 training-test splits. The improvement and error is reported in terms of the doubly robust rewards for consistency. In the Data column, Full corresponds to the full embeddings as extracted from Clinical Longformer, CH corresponds to the Representation from the Classification Head, and PCA to the embeddings after PCA reduction. Estimator refers to the reward estimation method, using either tabular features only or both tabular and full note embeddings (multimodal).

| Rewards | Data | TAVR models | Liver trauma models |
|---|---|---|---|
| Doubly robust | Tab | $79.06 \pm 2.23$ | $86.77 \pm 0.78$ |
| | Full | $81.68 \pm 3.03$ | $90.25 \pm 1.2$ |
| | CH | $92.47 \pm 3.14$ | $88.37 \pm 1.31$ |
| | PCA | $73.19 \pm 0.66$ | $85.30 \pm 0.85$ |
| Dragonet | Tab | $83.12 \pm 4.93$ | $85.74 \pm 2.49$ |
| | Full | $87.1 \pm 5.91$ | $87.37 \pm 2.39$ |
| | CH | $91.12 \pm 5.02$ | $84.33 \pm 2.61$ |
| | PCA | $84.14 \pm 5.1$ | $80.6 \pm 4.54$ |
| TARNet | Tab | $84.59 \pm 4.51$ | $88.37 \pm 2.74$ |
| | Full | $81.27 \pm 5.0$ | $86.99 \pm 3.0$ |
| | CH | $91.61 \pm 4.96$ | $87.16 \pm 2.08$ |
| | PCA | $81.14 \pm 5.34$ | $77.84 \pm 5.46$ |

### A.3.2 Structured Data

We present full results for the structured datasets under different reward estimation methods (doubly robust and Causal Forest) for Optimal Policy Trees and PNNs in Table 18 and the respective Mirrored OCT training accuracies in Table 19. The other methods discussed in the main text (Regress & Compare, Causal Forest) do not use rewards in their training algorithm. For consistency, improvements are reported on test set rewards estimated using the doubly robust method. We observe that under the causal forest rewards, PNNs and Mirrored OCTs result in comparable performance to Optimal Policy Trees, but achieve higher average outcome improvement in three out of the four datasets. This observation is consistent with the results discussed in the main text, demonstrating the ability of PNNs to perform well when trained with different types of rewards.

Table 18: Improvement (%) in estimated outcome rewards (test set) for the experiments with structured data under different reward estimation models. We report the average improvement and standard error across five random 50-50 training-test splits. The improvement and error is reported in terms of the doubly robust rewards for consistency.

| Rewards | Method | Diabetes | Groceries | Spleen | REBOA |
|---|---|---|---|---|---|
| Doubly robust | Optimal Policy Tree | $2.55 \pm 0.52$ | $106.58 \pm 2.38$ | $12.98 \pm 1.23$ | $17.17 \pm 3.68$ |
| | PNN | $\mathbf{3.15 \pm 0.51}$ | $\mathbf{110.88 \pm 1.18}$ | $\mathbf{13.52 \pm 1.74}$ | $17.88 \pm 3.88$ |
| | Mirrored OCT | $3.06 \pm 0.53$ | $110.22 \pm 6.94$ | $9.47 \pm 1.91$ | $\mathbf{18.09 \pm 3.18}$ |
| Causal Forest | Optimal Policy Tree | $1.82 \pm 0.51$ | $\mathbf{67.2 \pm 10.32}$ | $8.9 \pm 2.41$ | $4.0 \pm 3.46$ |
| | PNN | $\mathbf{1.90 \pm 0.47}$ | $65.22 \pm 7.37$ | $\mathbf{10.36 \pm 2.49}$ | $6.85 \pm 2.91$ |
| | Mirrored OCT | $1.83 \pm 0.47$ | $65.2 \pm 7.37$ | $7.26 \pm 2.05$ | $\mathbf{7.13 \pm 2.95}$ |

Table 19: Training accuracy (%) of the Mirrored OCTs for the structured datasets under different reward estimation models. We report the average accuracy and standard error across five random 50-50 training-test splits.

| Rewards | Diabetes | Groceries | Spleen | REBOA |
|---|---|---|---|---|
| Doubly robust | $86.09 \pm 1.35$ | $98.03 \pm 0.36$ | $92.76 \pm 0.86$ | $95.72 \pm 0.37$ |
| Causal Forest | $92.33 \pm 1.34$ | $100.0 \pm 0.00$ | $88.41 \pm 2.27$ | $96.23 \pm 1.61$ |

## A.4 Uncertainty of Estimated Outcome Rewards

This section quantifies the uncertainty of estimated outcome rewards for each method trained on our unstructured and structured datasets. Since we have trained a separate rewards model on the test set as discussed in the main text, we have used these estimated test set rewards to perform the bootstrapping procedure in this section of the appendix. We perform a nonparametric bootstrap with 500 resamples of respective test sets from each 50-50 split and using the trained models of the main text. From this procedure, we report the policy value and 95% confidence interval, both measured in terms of improvement as a percentage.

### A.4.1 Unstructured Data

We present the uncertainty for the of estimated outcome rewards for the unstructured datasets. We observe that although improvements evaluated on the rewards from the tabular model are more modest, the results on the TAVR dataset are quite stable; in the liver injury dataset, the wider confidence intervals are attributed to the small dataset size and the relatively smaller overlap between treatment groups.

Table 20: Policy value and 95% confidence interval from a nonparametric bootstrap of the test sets from the 50-50 splits, measured in terms of improvement (%), for the unstructured datasets. Seed refers to the particular seeded 50-50 split and estimator refers to the reward estimation method, using either tabular features only or both tabular and full note embeddings (multimodal). "Tabular" refers to the tabular model and "multimodal" refers to the multimodal model. For each seed, estimator, and method, we present the uncertainty for one model instance.

| Seed | Estimator | Method | TAVR Tabular | TAVR Multimodal | Liver trauma Tabular | Liver trauma Multimodal |
|---|---|---|---|---|---|---|
| 1 | Tabular | PNN | 2.91 [−12.31, 18.63] | 6.99 [−11.58, 25.22] | 19.37 [−17.43, 47.99] | 41.99 [13.31, 69.7] |
| | | Mirrored OCT | 2.35 [−15.09, 19.26] | 3.88 [−15.96, 22.11] | 39.01 [7.6, 64.88] | 39.01 [7.6, 64.88] |
| | Tabular & Notes | PNN | 16.86 [3.96, 31.42] | 16.92 [3.18, 30.48] | 26.47 [−5.81, 52.89] | 37.58 [8.45, 63.45] |
| | | Mirrored OCT | 11.36 [−4.16, 25.78] | 8.01 [−7.54, 23.16] | 38.81 [8.2, 65.01] | 38.81 [8.2, 65.01] |
| 2 | Tabular | PNN | 14.62 [1.28, 27.3] | 35.1 [27.77, 42.33] | 16.73 [−17.23, 47.97] | 30.94 [−1.57, 61.75] |
| | | Mirrored OCT | 14.56 [0.51, 27.43] | 35.04 [27.77, 42.27] | 21.77 [−8.78, 50.88] | 12.28 [−21.12, 41.52] |
| | Tabular & Notes | PNN | 24.32 [10.37, 35.65] | 55.41 [47.74, 62.01] | 24.47 [−5.37, 51.65] | 22.83 [−11.7, 53.53] |
| | | Mirrored OCT | 23.12 [9.13, 35.38] | 55.32 [47.66, 61.88] | 25.25 [−5.58, 53.69] | 17.11 [−15.09, 46.54] |
| 3 | Tabular | PNN | 7.04 [−7.63, 21.36] | 30.42 [22.08, 37.9] | 4.57 [−29.82, 38.03] | 21.94 [−11.54, 54.53] |
| | | Mirrored OCT | 7.28 [−6.97, 21.79] | 32.99 [25.72, 40.75] | 25.92 [−6.4, 55.62] | 6.47 [−29.9, 40.54] |
| | Tabular & Notes | PNN | 12.55 [−0.78, 27.03] | 46.09 [39.36, 52.48] | 18.69 [−10.36, 46.63] | 23.49 [−8.56, 55.13] |
| | | Mirrored OCT | 7.22 [−7.41, 20.86] | 46.62 [40.04, 52.28] | 30.27 [2.33, 57.91] | 13.96 [−24.21, 48.2] |
| 4 | Tabular | PNN | 5.99 [−8.33, 18.6] | 20.54 [10.52, 29.9] | 18.93 [−13.54, 51.39] | 20.3 [−12.43, 49.22] |
| | | Mirrored OCT | 9.87 [−3.83, 22.59] | 20.12 [9.34, 30.27] | 32.16 [0.37, 59.32] | 40.61 [12.01, 68.26] |
| | Tabular & Notes | PNN | 23.19 [11.12, 35.36] | 46.65 [37.94, 54.99] | 23.43 [−4.28, 51.56] | 21.87 [−9.01, 51.04] |
| | | Mirrored OCT | 26.95 [14.43, 38.27] | 46.47 [38.08, 54.66] | 38.09 [10.77, 63.21] | 31.66 [−0.69, 59.4] |
| 5 | Tabular | PNN | −10.92 [−26.88, 5.49] | −4.98 [−10.09, −0.16] | 11.85 [−24.95, 44.13] | 21.35 [−11.02, 52.95] |
| | | Mirrored OCT | −11.02 [−27.08, 5.71] | −2.65 [−6.96, 1.26] | 40.12 [11.29, 65.03] | 19.9 [−11.41, 51.21] |
| | Tabular & Notes | PNN | 22.98 [10.79, 36.11] | 50.45 [43.33, 58.35] | 26.28 [−4.4, 51.67] | 30.48 [−2.83, 57.42] |
| | | Mirrored OCT | 23.07 [11.02, 36.35] | 51.66 [44.78, 58.98] | 41.96 [13.49, 66.52] | 18.52 [−12.44, 48.06] |

### A.4.2 Structured Data

We present the uncertainty for estimated outcome rewards for the structured datasets. We observe that the groceries and diabetes datasets are the ones with the narrower confidence intervals; for groceries there is also no overlap between PNN or Mirrored OCT and next best model improvements in four out of five seeds. Although the REBOA dataset has the smallest positivity overlap, the results are quite robust, partially increasing our confidence in their interpretation.

Table 21: Policy value and 95% confidence interval from a nonparametric bootstrap of the 50-50 split test set, measured in terms of improvement (%), for the structured datasets. Seed refers to the particular seeded 50-50 split. "Tabular" refers to the tabular model and "multimodal" refers to the multimodal model. For each seed and method, we present the uncertainty for one model instance.

| Seed | Method | Diabetes | Groceries | Spleen | REBOA |
|---|---|---|---|---|---|
| 1 | Regress & Compare | $-1.05$ $[-2.9, 0.56]$ | $93.29$ $[92.13, 94.53]$ | $13.86$ $[8.41, 19.76]$ | $-1.04$ $[-6.9, 4.48]$ |
|  | Causal Forest | $-0.61$ $[-2.57, 0.97]$ | $104.6$ $[103.28, 105.95]$ | $14.75$ $[3.23, 25.33]$ | $-18.88$ $[-25.89, -12.35]$ |
|  | OPT | $3.44$ $[1.95, 5.5]$ | $113.77$ $[112.12, 115.43]$ | $16.52$ $[5.19, 27.59]$ | $23.48$ $[17.25, 29.02]$ |
|  | PNN | $2.9$ $[2.1, 3.97]$ | $119.6$ $[118.04, 121.21]$ | $18.21$ $[7.81, 28.24]$ | $25.83$ $[18.6, 32.87]$ |
|  | Mirrored OCT | $2.7$ $[2.0, 3.61]$ | $116.48$ $[115.03, 117.98]$ | $13.82$ $[3.84, 23.17]$ | $26.27$ $[21.01, 30.9]$ |
| 2 | Regress & Compare | $5.2$ $[4.89, 5.56]$ | $86.14$ $[84.87, 87.33]$ | $-87.03$ $[-103.31, -70.88]$ | $0.33$ $[-5.15, 6.42]$ |
|  | Causal Forest | $3.6$ $[3.11, 4.33]$ | $84.8$ $[83.44, 86.15]$ | $-18.85$ $[-34.16, -5.29]$ | $-14.82$ $[-21.37, -8.71]$ |
|  | OPT | $4.19$ $[3.52, 4.85]$ | $100.49$ $[98.58, 102.2]$ | $8.99$ $[-2.22, 20.64]$ | $13.53$ $[6.56, 19.16]$ |
|  | PNN | $5.14$ $[4.39, 5.98]$ | $97.58$ $[95.89, 99.39]$ | $8.49$ $[-5.37, 19.79]$ | $15.68$ $[8.78, 21.86]$ |
|  | Mirrored OCT | $5.34$ $[4.52, 6.32]$ | $96.16$ $[94.59, 98.01]$ | $7.05$ $[-4.87, 18.02]$ | $18.45$ $[12.66, 24.08]$ |
| 3 | Regress & Compare | $2.61$ $[2.41, 2.82]$ | $100.54$ $[99.26, 101.82]$ | $5.21$ $[-0.9, 10.73]$ | $-1.95$ $[-7.75, 4.5]$ |
|  | Causal Forest | $0.12$ $[-1.53, 1.89]$ | $102.3$ $[100.73, 103.74]$ | $-5.56$ $[-17.61, 7.78]$ | $-11.08$ $[-18.36, -3.29]$ |
|  | OPT | $0.75$ $[-0.79, 2.13]$ | $107.85$ $[106.22, 109.32]$ | $8.76$ $[-2.47, 18.93]$ | $22.8$ $[18.07, 27.68]$ |
|  | PNN | $1.82$ $[0.06, 3.98]$ | $118.86$ $[117.05, 120.79]$ | $10.76$ $[-0.19, 21.06]$ | $22.79$ $[16.53, 28.81]$ |
|  | Mirrored OCT | $1.1$ $[-0.47, 3.11]$ | $117.91$ $[116.13, 119.8]$ | $4.21$ $[-6.32, 14.67]$ | $22.8$ $[18.07, 27.68]$ |
| 4 | Regress & Compare | $2.37$ $[2.05, 2.8]$ | $89.64$ $[88.34, 90.8]$ | $-22.42$ $[-37.48, -7.75]$ | $1.05$ $[-4.73, 6.45]$ |
|  | Causal Forest | $1.81$ $[0.55, 3.49]$ | $99.43$ $[97.88, 100.92]$ | $8.07$ $[-4.93, 20.02]$ | $-37.52$ $[-44.56, -30.15]$ |
|  | OPT | $1.89$ $[0.71, 3.13]$ | $106.17$ $[104.42, 108.08]$ | $12.17$ $[1.06, 22.33]$ | $6.44$ $[0.96, 11.43]$ |
|  | PNN | $3.62$ $[2.16, 5.79]$ | $117.75$ $[115.73, 119.88]$ | $12.96$ $[2.27, 22.65]$ | $4.06$ $[-1.92, 9.1]$ |
|  | Mirrored OCT | $3.17$ $[1.56, 5.35]$ | $115.8$ $[113.88, 117.73]$ | $16.09$ $[6.5, 23.87]$ | $6.89$ $[1.21, 12.1]$ |
| 5 | Regress & Compare | $2.97$ $[2.76, 3.18]$ | $95.07$ $[93.78, 96.16]$ | $-114.28$ $[-132.72, -97.28]$ | $2.17$ $[-3.54, 7.93]$ |
|  | Causal Forest | $1.06$ $[0.52, 1.48]$ | $101.58$ $[99.96, 102.92]$ | $-3.55$ $[-16.39, 11.3]$ | $-14.54$ $[-23.93, -5.7]$ |
|  | OPT | $1.55$ $[0.15, 2.65]$ | $103.65$ $[101.9, 105.34]$ | $16.17$ $[5.01, 25.91]$ | $20.39$ $[14.7, 26.08]$ |
|  | PNN | $2.8$ $[1.93, 3.82]$ | $113.68$ $[111.67, 115.49]$ | $18.48$ $[7.58, 29.39]$ | $19.62$ $[13.16, 25.1]$ |
|  | Mirrored OCT | $2.93$ $[2.05, 3.92]$ | $113.5$ $[111.55, 115.35]$ | $12.06$ $[0.54, 22.63]$ | $21.3$ $[15.73, 26.75]$ |

### A.5 PNN Hyperparameters

We specify and tune the following PNN hyperparameters:

- **Number of layers of the network.** We experiment with both shallow and deep networks. Though conclusions differ based on the dataset, in general we observe that deeper networks do not necessarily improve results.

- **Number of nodes at each layer.** The size of the dataset closely affects this hyperparameter. Typically more nodes per layer are used with larger datasets that also include more features.

- **Batch size.** This parameter determines the number of samples used in each forward pass of the network and for backpropagation, where the network parameters are updated after each batch passes through the network. It also affects the training speed, since too many batches can slow down the training process. Again, there is a correlation between batch size and the size of our data; larger batch sizes are employed for larger datasets.

- **Learning rate.** The learning rate is an important parameter of the training process, since it defines how steep the descent is at each step of the gradient descent algorithm during training. After experimentation for each dataset, we find an appropriate learning rate that is not too big so that the algorithm becomes stuck in local optima but also not too small so that convergence is too slow.

- **Weight decay.** This hyperparameter scales an $L_2$-regularization term of the network weights that is added to the objective function to prevent them from taking too large values. Since our data is normalized, we observe that lowering the weight decay coefficient and relaxing the weights are actually beneficial and do not result in overfitting.

- **Number of epochs.** The number of epochs is a particularly hard parameter to tune, since we want to prevent overfitting but also allow the training to continue until sufficient convergence. For this reason, we employ early stopping, a technique that is adaptive to each specific training process and terminates training when a fluctuation in the validation loss is observed. Such a fluctuation indicates that the network is no longer improving in out-of-sample data generalization.

## A.6 Computational Analysis

Table 22: Computational training time (in seconds) for each model and unstructured dataset on the 50/50 split. Times reported are averaged over 5 random seeds. PNNs were trained on a single NVIDIA Volta-class GPU (xeon-g6-volta compute node). Mirrored OCT models were trained on CPU.

| Dataset | # Train Samples | Embedding | PNN | Mirrored OCT |
|---------|-----------------|-----------|-----|--------------|
| TAVR | 903 | Tab | 45.27 | **2.44** |
| | | Full | 36.39 | **24.19** |
| | | CH | 30.58 | **16.91** |
| | | PCA | 43.13 | **13.64** |
| Liver | 323 | Tab | 27.49 | **1.28** |
| | | Full | 19.83 | **4.52** |
| | | CH | 16.24 | **3.34** |
| | | PCA | 14.48 | **2.75** |

Table 23: Computational training time (in seconds) for each model and structured dataset on the 50/50 split. Times reported are averaged over 5 random seeds. PNNs were trained on a single NVIDIA Volta-class GPU (xeon-g6-volta compute node). All other models were trained on CPU.

| Dataset | # Train Samples | Regress & Compare | Causal Forest | OPT | PNN | Mirrored OCT |
|---------|-----------------|-------------------|---------------|-----|-----|--------------|
| Diabetes | 19,992 | **15.10** | 176.19 | 170.75 | 330.05 | 80.09 |
| Groceries | 41,351 | **14.12** | 26.74 | 78.27 | 684.00 | 52.48 |
| Spleen | 13,749 | **14.01** | 37.09 | 35.47 | 225.02 | 32.69 |
| REBOA | 3,822 | 13.37 | 11.83 | 10.82 | 74.36 | **6.38** |

Across structured datasets, Regress & Compare is consistently one of the fastest methods. This is expected: the approach reduces prescription to fitting a single supervised learning model (here, XGBoost), which is highly optimized, parallelized, and scales approximately linearly with the number of samples.

The remaining patterns reflect the inherent computational characteristics of each model family. Causal Forests aggregate hundreds of trees and therefore are associated with higher training cost. OPT and Mirrored OCT benefit from the depth constraint we use (depth = 5), which keeps the combinatorial search space manageable; training time would increase rapidly with deeper trees. PNNs require many forward and backward passes, so their cost scales approximately linearly with the number of parameters, samples, and epochs, leading to higher runtimes, especially when the number of samples is large.

Overall, the efficiency ranking observed in Tables 22–23 mirrors well-known theoretical scaling properties: XGBoost-based Regress & Compare is the fastest, tree-based methods land in the middle depending on

depth, and neural networks incur the highest cost due to iterative gradient optimization. However, the difference between PNNs and trees decreases, if one chooses to train deeper trees.

## A.7 Additional tables

Table 24: Treatment scenarios, data splits, and counterfactual estimation methods used for each structured dataset.

| Dataset | Treatment scenario | Counterfactual estimator |
|---|---|---|
| Groceries pricing | Single continuous | XGBoost Classifier (direct method) |
| Splenic injuries treatment | Multiple discrete | Random Forest Classifier (doubly robust) |
| REBOA in blunt trauma patients | Binary | Random Forest Classifier (doubly robust) |
| Diabetes management | Multiple continuous | Random Forest Regressor (doubly robust) |

## A.8 AOs before and after trimming

Figure 8: AOs before and after trimming.

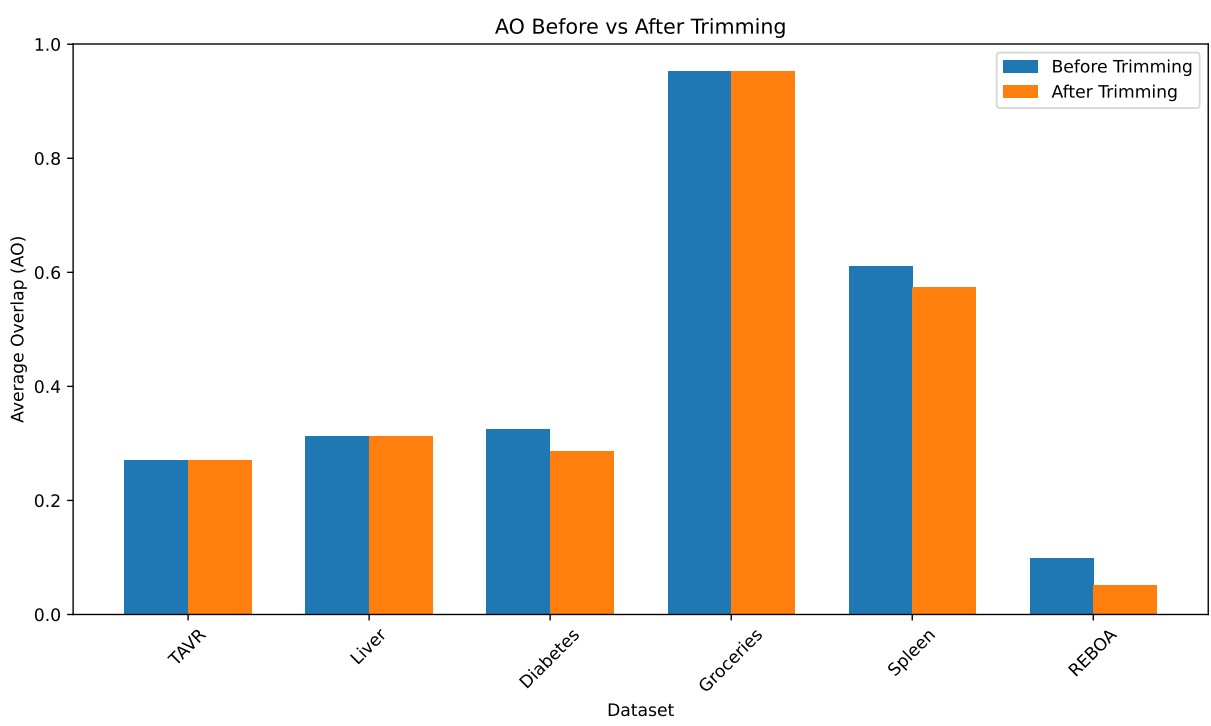

## A.9 Positivity plots before trimming

Figure 9: TAVR propensity score distributions.

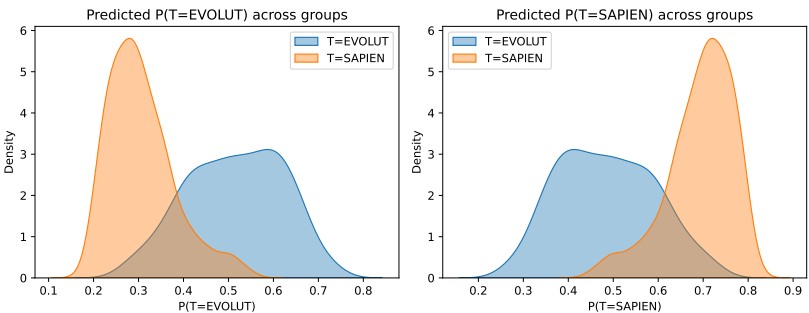

Figure 10: Liver injury propensity score distributions.

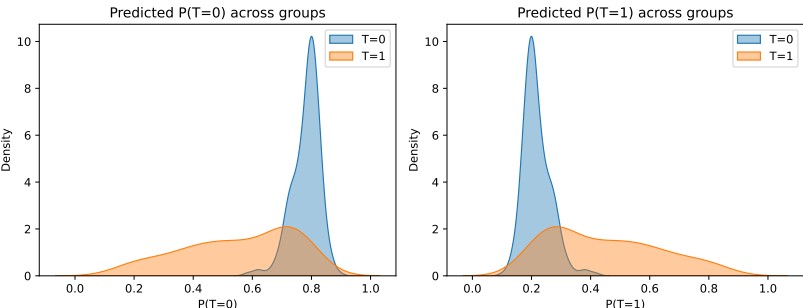

Figure 11: Diabetes propensity score distributions before trimming.

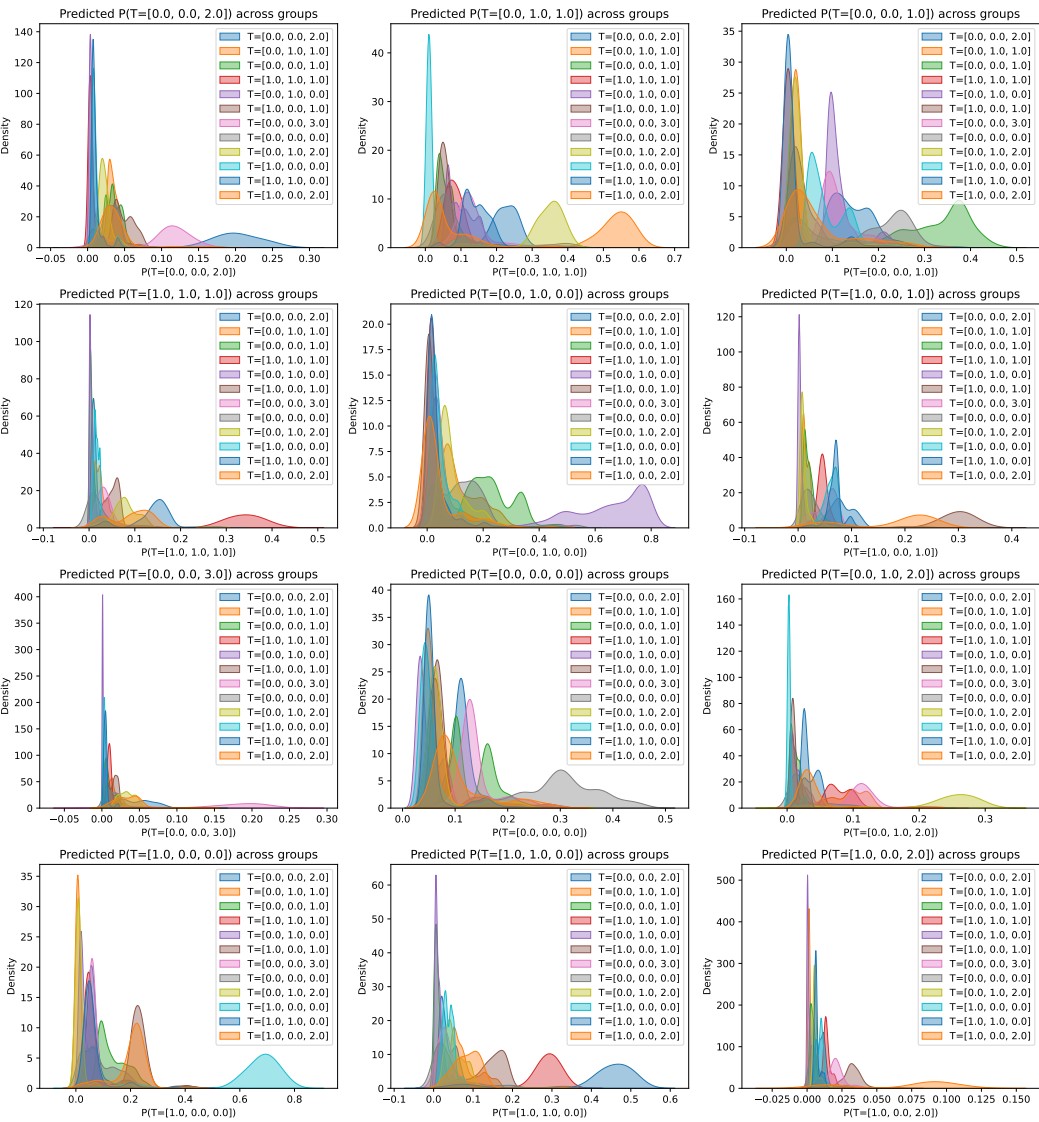

Figure 12: Groceries propensity score distributions before trimming.

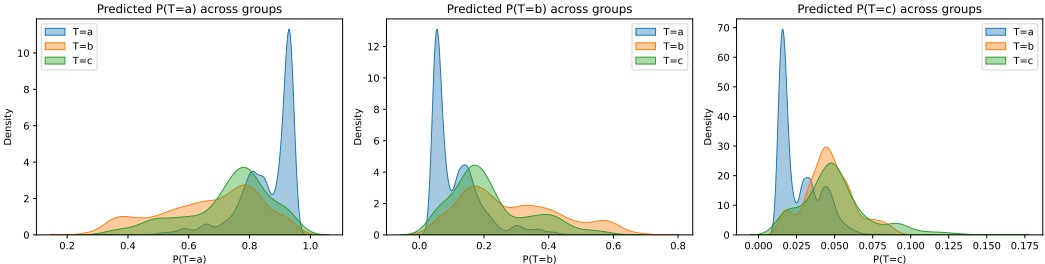

Figure 13: Spleen propensity score distributions before trimming.

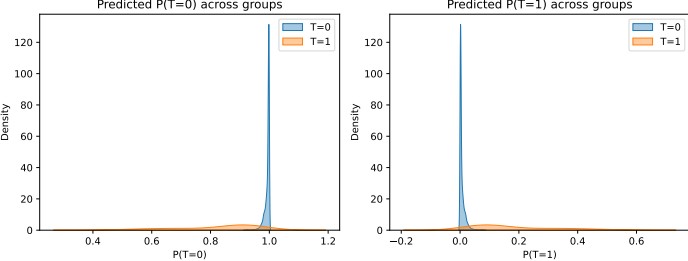

Figure 14: REBOA propensity score distributions before trimming.

## A.10 Positivity plots after trimming

Figure 15: Diabetes propensity score distributions after trimming.

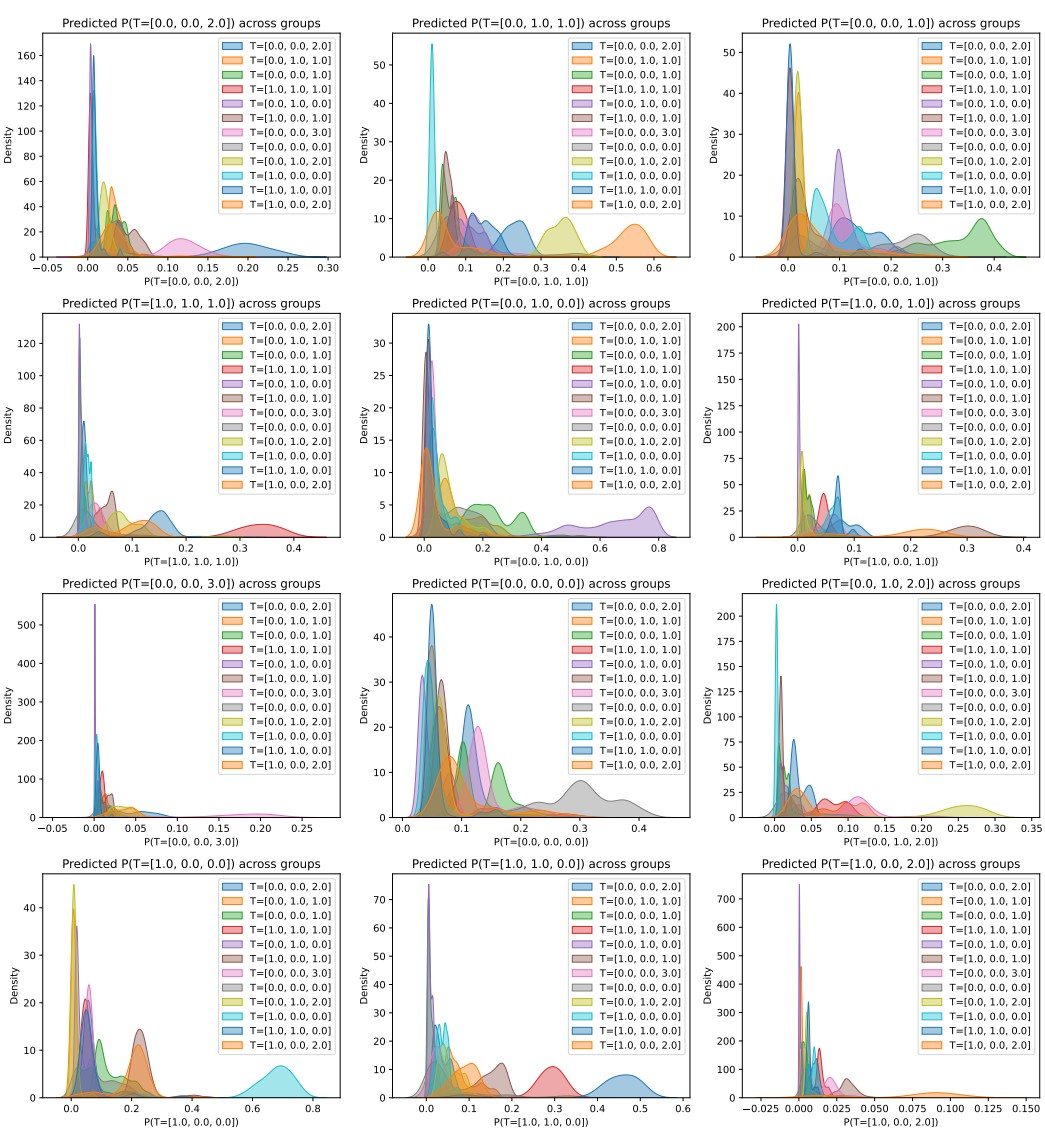

Figure 16: Spleen propensity score distributions after trimming.

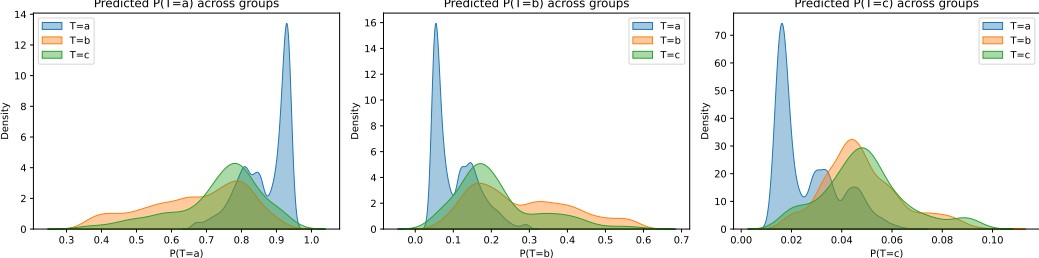

Figure 17: REBOA propensity score distributions after trimming.

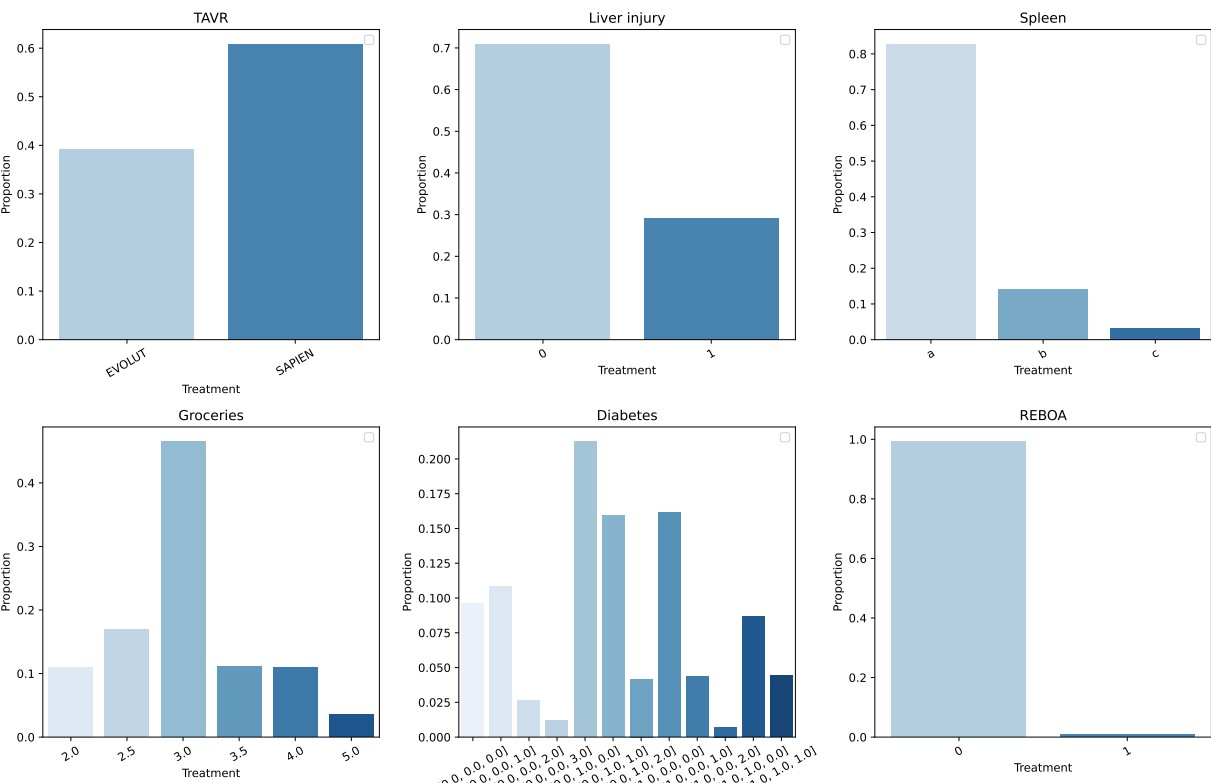

## A.11 Treatment distribution after trimming

Figure 18: Treatment distribution grid after trimming.

The treatments for the different datasets are the following:

- **TAVR**: There are two potential valves, SAPIEN and EVOLUT.

- **Liver injury**: 0 corresponds to no surgery, and 1 corresponds to prescribing surgery.

- **Diabetes**: Each vector contains the dosage of insulin, metformin and oral treatment in that order.

- **Groceries**: Each value corresponds to the prescribed price, in dollars.

- **Spleen**: Treatment "a" corresponds to observation, treatment "b" corresponds to splenectomy and treatment "c" corresponds to angioembolization.

- **REBOA**: 0 corresponds to no procedure, and 1 corresponds to prescribing the procedure.

## A.12   SMDs before and after trimming

Figure 19: Average SMDs across datasets before and after trimming.

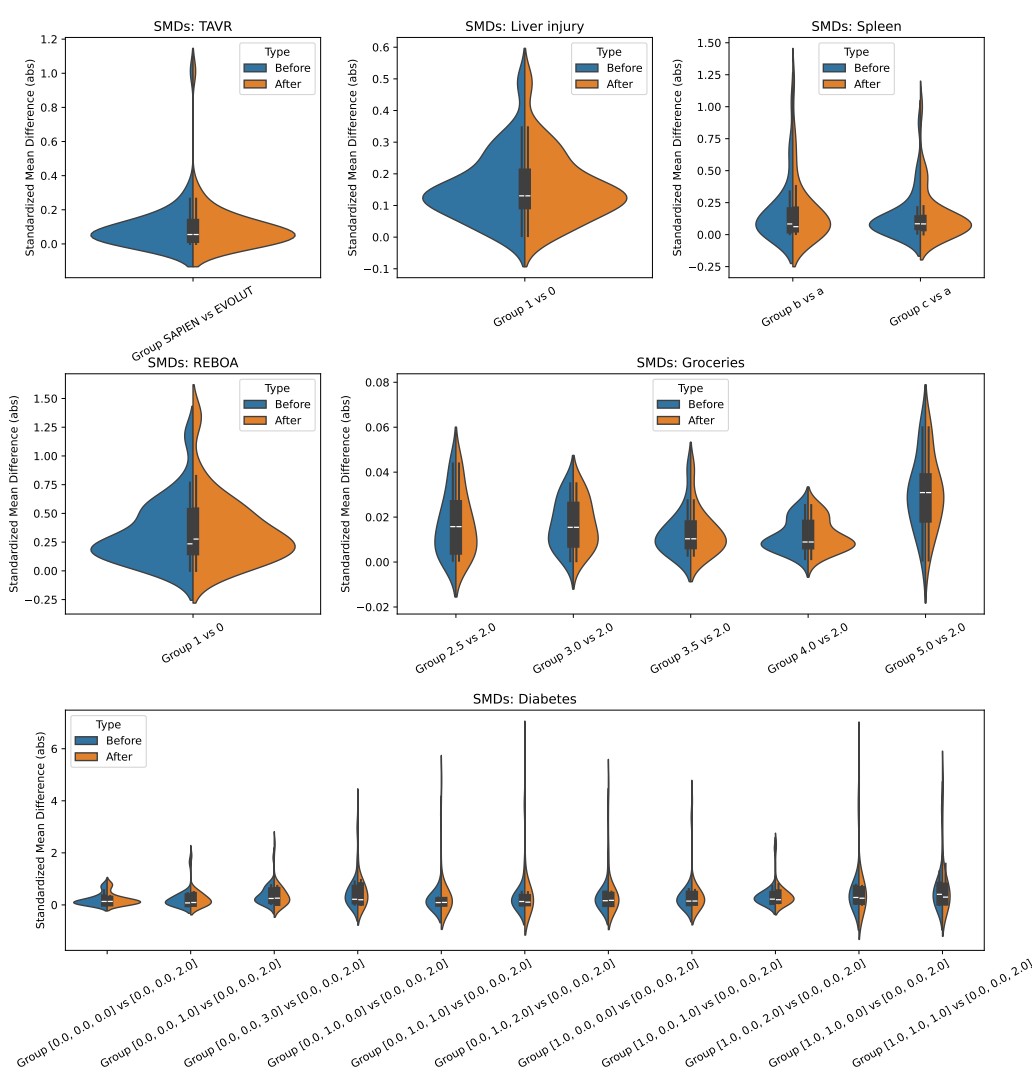

## A.13   Unstructured datasets: Notes analysis

In this section, we discuss an analysis of the notes data used in the unstructured datasets, which confirm the absence of both temporal and content-based label leakage.

### A.13.1 TAVR

To ensure that there is no data leakage from either treatment labels or outcomes, we restrict the corpus to notes recorded strictly before the date of the TAVR procedure (at least one day prior). Figure 20 shows the distribution of note timestamps relative to the day of the TAVR procedure. Most notes precede the procedure by fewer than 70 days, confirming that text information originates from the recent preoperative period.

Figure 20: Distribution of note timestamps relative to the TAVR procedure date. Negative values correspond to notes written before the procedure.

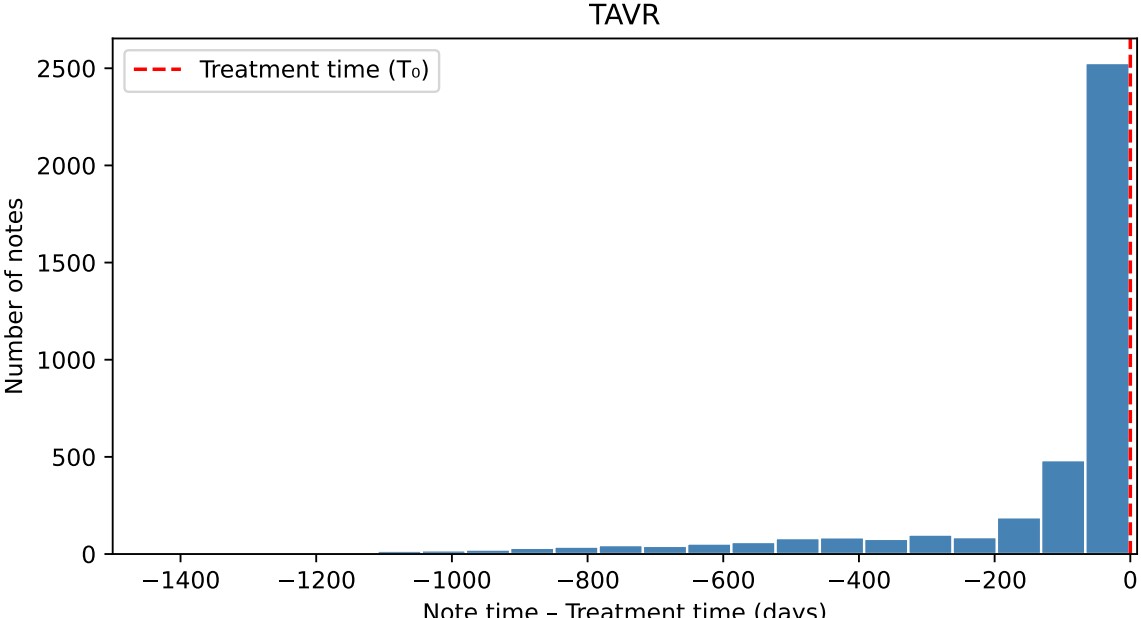

Even though all notes were produced prior to treatment, we further verify that no textual content reveals future information. We search for treatment-related keywords (*Evolut, Sapien*) and outcome-related terms (*pacemaker, implantation, PPI*). None of the treatment keywords appeared in the notes. Mentions of *pacemaker* or *implantation* refer exclusively to pre-existing devices visible in imaging reports or to general procedural risk discussions.

### A.13.2 Liver trauma

We note that the cohort of the liver trauma dataset is restricted to patients who had imaging done during their hospital admission, and we received the dataset after some external preprocessing. In particular, the notes for each patient in the dataset are a combined mixture of various relevant notes, with no fixed date, and importantly, these notes occur strictly prior to the timestamp of each patient's imaging. Since notes occur before preoperative imaging, these notes cannot contain any information about the outcome, patient mortality.

We additionally verify that the texts do not leak information about treatment labels. We search for treatment-related keywords, which includes the specific surgeries we included as valid candidates for the liver surgery treatment, and none of these keywords appeared in the notes.

### A.14 Extension of interpretability of unstructured and structured real-world datasets

**Liver trauma.** We discuss the multimodal Mirrored OCT from Figure 22 for the liver injury dataset. The embedding features from the clinical notes are not nearly as interpretable, but we can still interpret the tabular features selected by the Mirrored OCT. We see that the OCT selects two note features, as well as whether the patient had other symptoms with the circulatory or respiratory system and whether the patient had pain in their throat or chest.

**Groceries.** We compare the resulting Mirrored OCT from Figure 25 and the Optimal Policy Tree (OPT) from Figure 26. The features selected by both the Mirrored OCT and the Optimal Policy Tree are similar, with the most prominent ones being the homeowner status, age, income range, household status, and marital status. An interesting difference between the two trees is the number of distinct prescriptions selected: the OCT only selects 3 of the 6 pricing options – prediction classes 0, 1, and 5 (which correspond to prices USD $2.00, $2.50, and $5.00) – whereas the OPT prescribes more of the possible options. The OCT's strategy seems to therefore select low prices for lower-income households, compared to high prices for households with more financial stability.

**Splenic injuries treatment.** We compare the Mirrored OCT displayed in Figure 27 with an Optimal Policy Tree (OPT) in Figure 28. There are three possible treatments: simple observation (treatment 0 of the OCT, "a" for the OPT), splenectomy (treatment 1 of the OCT, "b" for the OPT) and angioembolization (treatment 2 of the OCT, "c" for the OPT). Both trees prescribe the first and third options. The two trees split on similar features, although at different levels of the tree; these features include SBP (systolic blood pressure), Age, and TBI (traumatic brain injury). However, the Mirrored OCT appears to be much deeper and utilizes more features, like e.g. pulse oximetry, therefore leveraging more aspects of the patients' clinical image.

**REBOA in blunt trauma patents.** We compare the Mirrored OCT from Figure 29 and the Optimal Policy Tree (OPT) from Figure 30.

From Figure 29, we clearly see that the important features include, among others, "gcs" (Glasgow coma scale), SBP (systolic blood pressure), and whether the patient is intubated. We observe that the OPT (Figure 30) splits on more features, that include, among others, the age and pulse rate of the patient.

### A.15 Mirrored OCTs on all real-world datasets

Further examples of Mirrored OCTs of maximum depth 7 for the TAVR, liver trauma, diabetes, groceries, and splenic injuries can be found under `https://drive.google.com/drive/folders/12XNOQllyzVQEFFguHvkcAQ-7Psp9KriQ`.

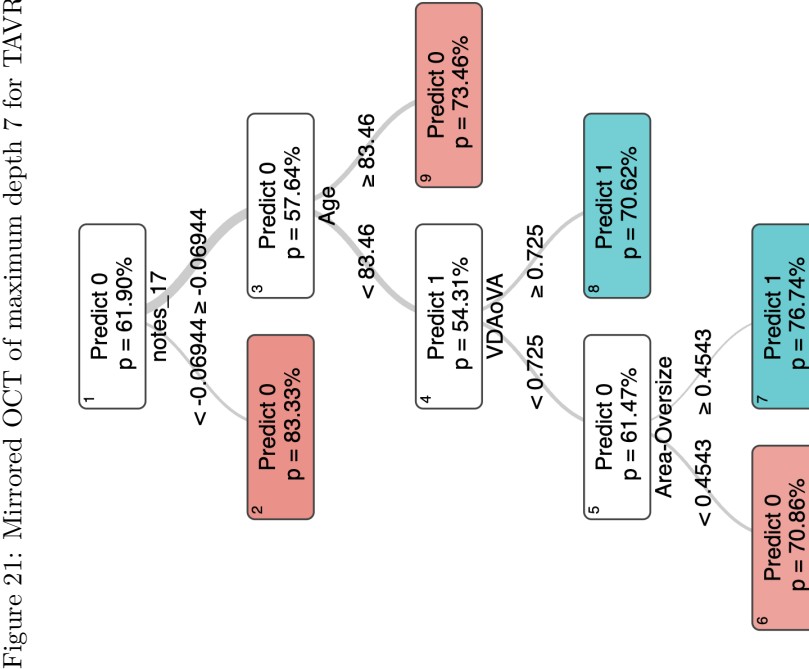

Figure 21: Mirrored OCT of maximum depth 7 for TAVR.

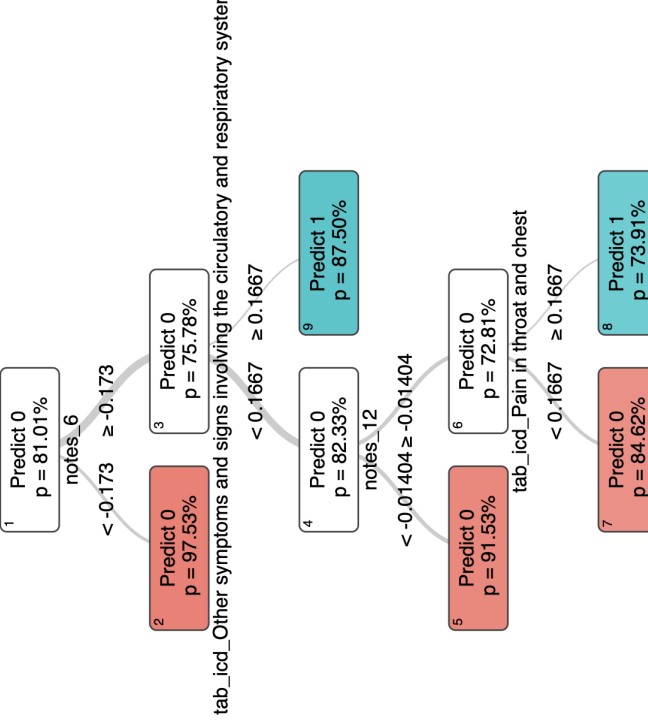

Figure 22: Mirrored OCT of maximum depth 7 for liver trauma.

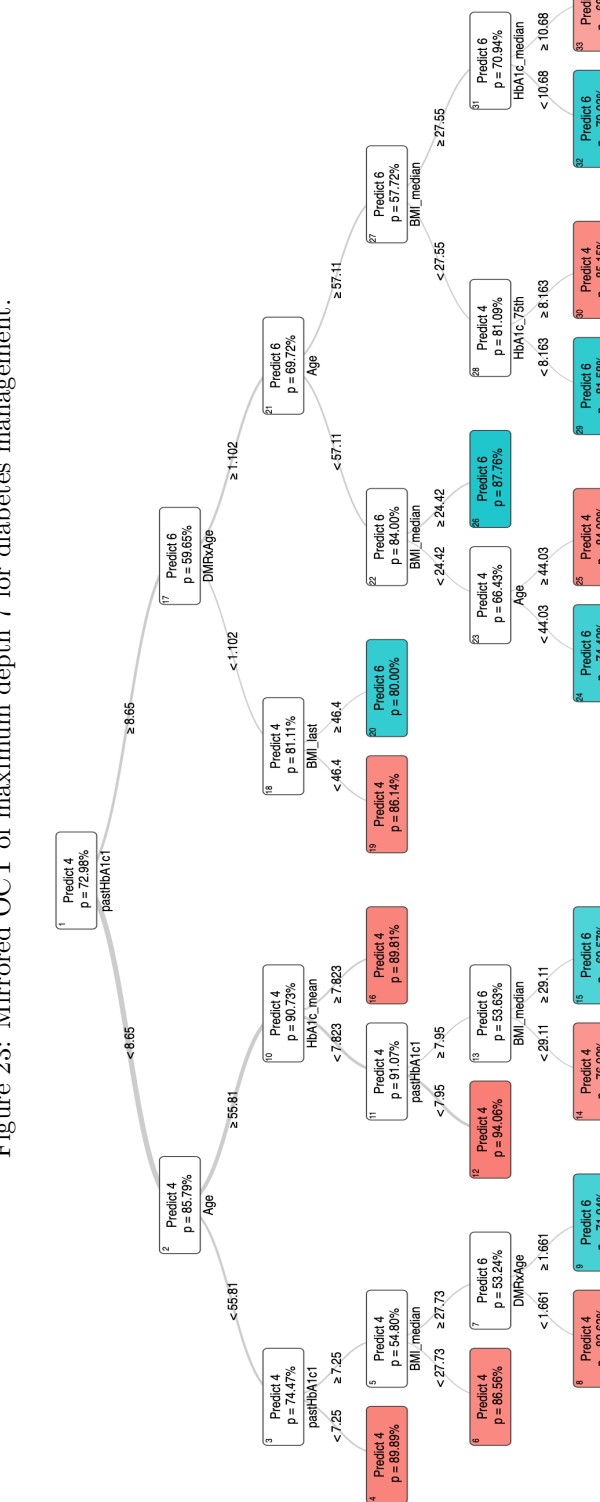

Figure 23: Mirrored OCT of maximum depth 7 for diabetes management.

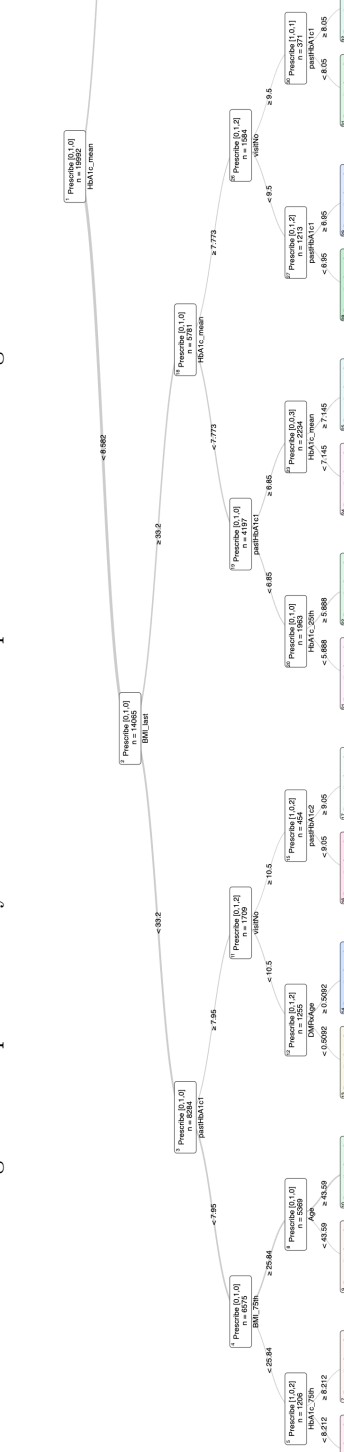

Figure 24: Optimal Policy Tree of maximum depth 7 for diabetes management.

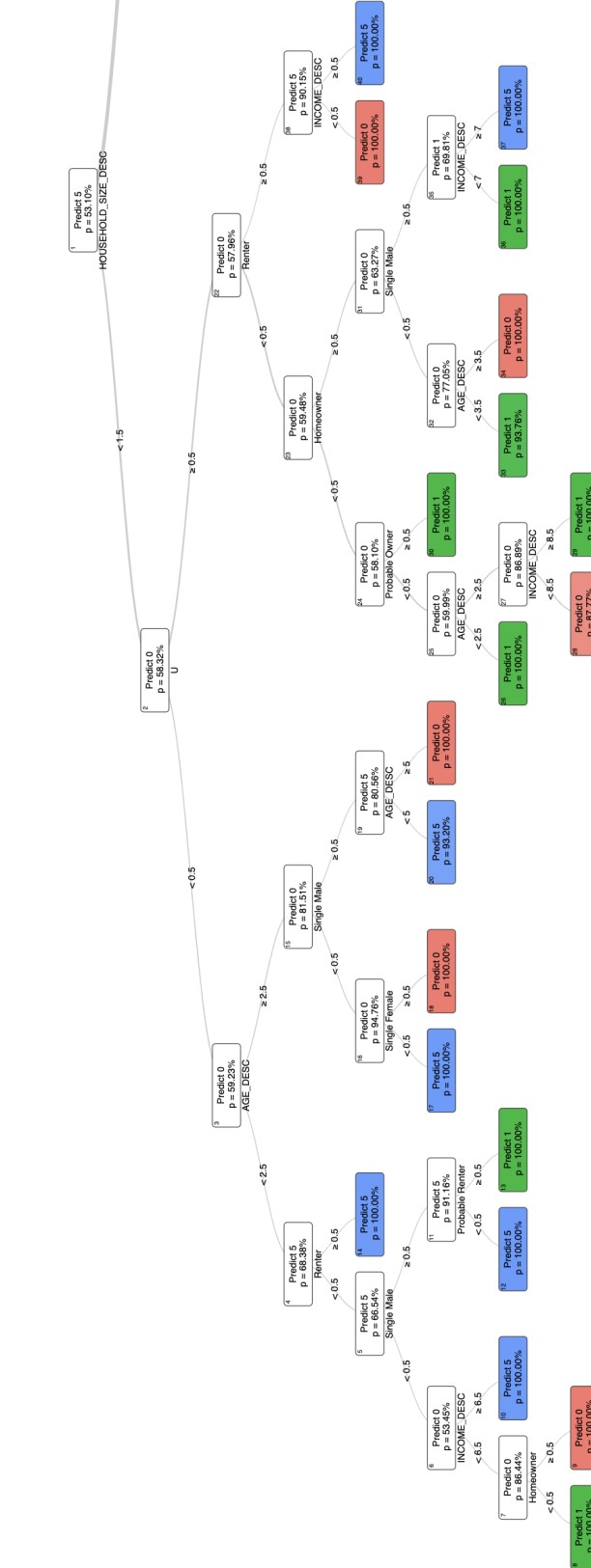

Figure 25: Mirrored OCT of maximum depth 7 for groceries.

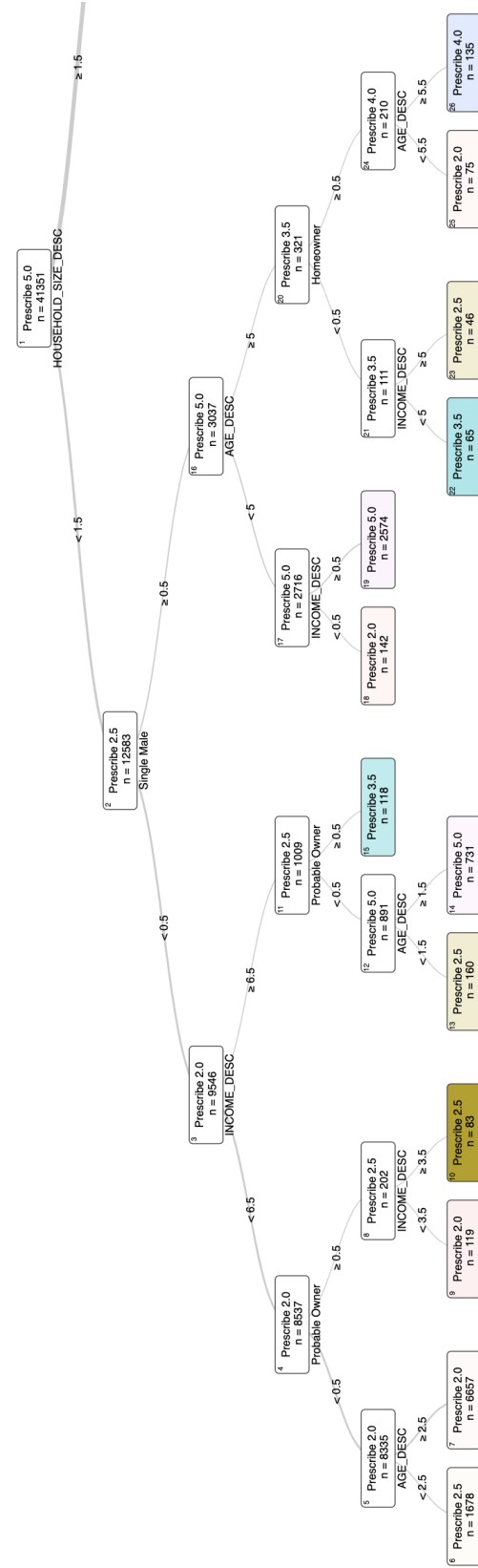

Figure 26: Optimal Policy Tree of maximum depth 7 for groceries.

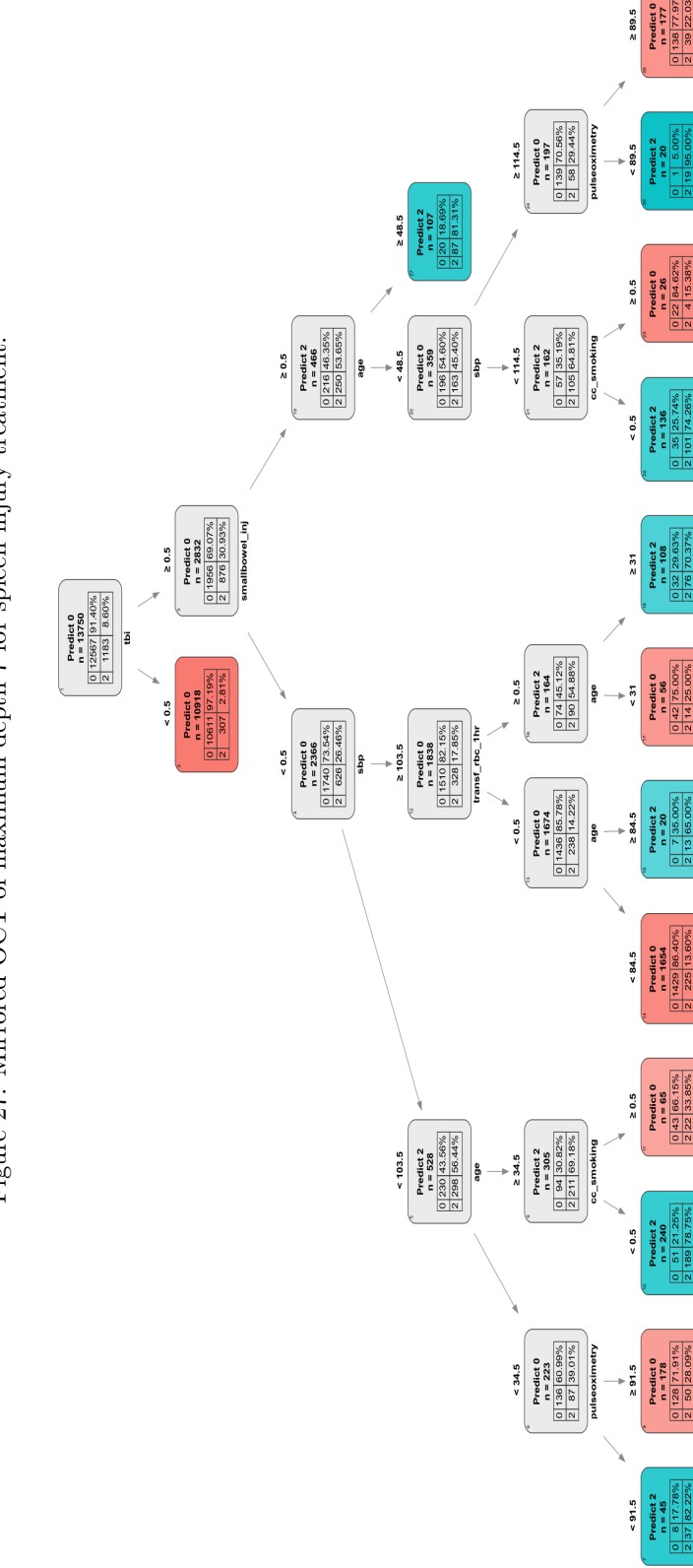

Figure 27: Mirrored OCT of maximum depth 7 for spleen injury treatment.

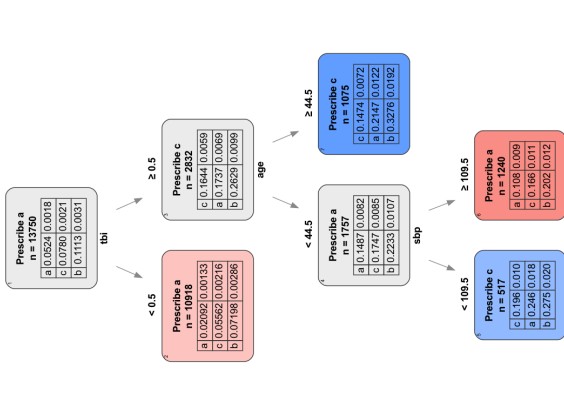

Figure 28: Optimal Policy Tree of maximum depth 7 for spleen injury treatment.

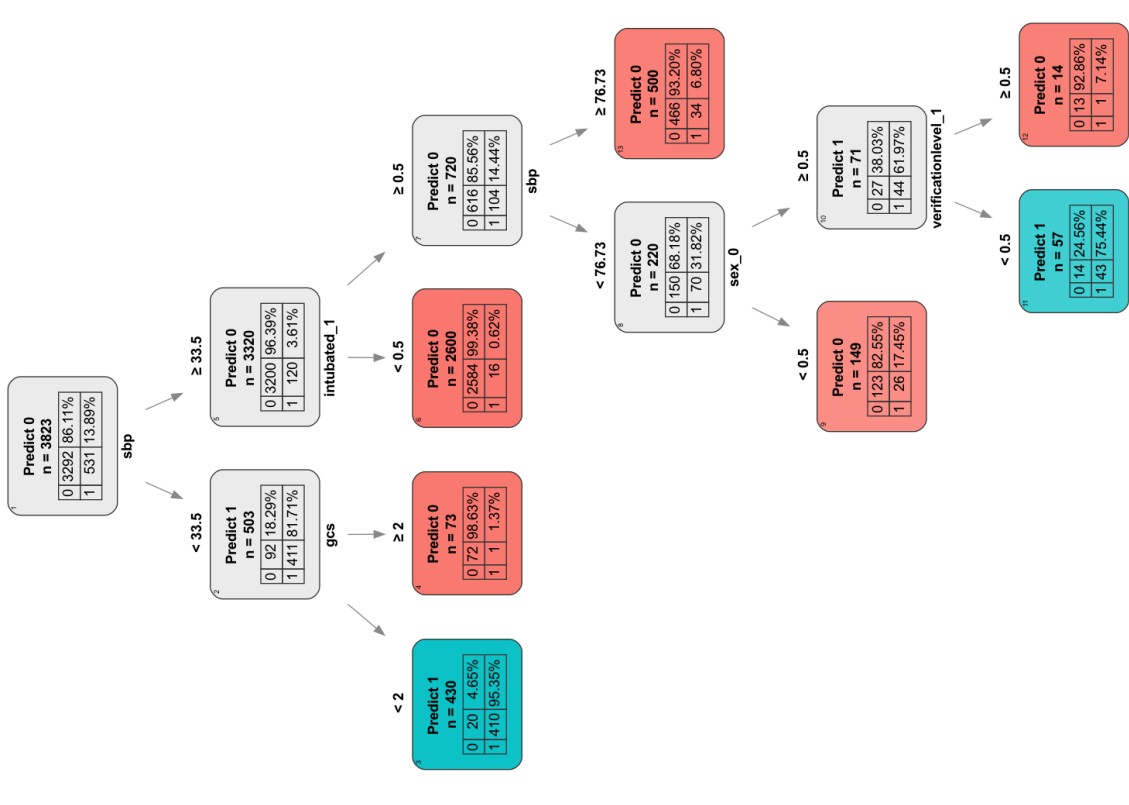

Figure 29: Mirrored OCT of maximum depth 7 for REBOA in blunt trauma patients.

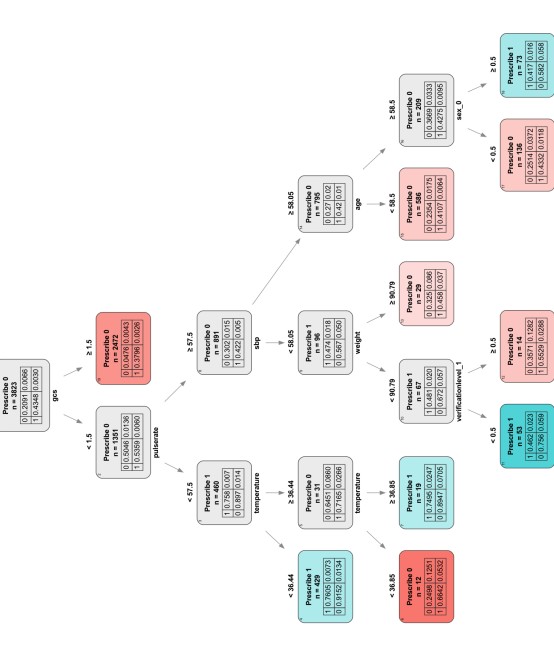

Figure 30: Optimal Policy Tree of maximum depth 7 for REBOA in blunt trauma patients.

