# OpenReview forum: "Multimodal Prescriptive Deep Learning"
_TMLR — Accepted by TMLR_

### Review · Reviewer_g9eQ · 2025-09-23

**Summary Of Contributions:**

The paper proposes Prescriptive Neural Networks (PNNs). The rewards are built by a counterfactual estimation step (direct method or doubly‑robust (DR) estimator, Eq. (1)), and the final prescription is the argmax treatment probability. For tabular datasets, the authors distill PNN prescriptions into Mirrored Optimal Classification Trees (OCTs) for interpretability. Experiments span two multimodal medical datasets (TAVR, liver trauma) and four tabular datasets (diabetes with multiple continuous treatments, groceries pricing with single continuous treatment, spleen with multiple discrete treatments, and REBOA with binary treatment).

**Audience:**

Yes

**Audience Explanation:**

The paper tackles multimodal prescription learning with a simple, scalable objective and provides a practical pathway to interpretability via Mirrored OCTs. The breadth of datasets and the unification across binary/multi‑discrete/continuous actions are of clear interest to readers working at the intersection of causal ML, deep learning, and policy optimization, even if several claims require tightening.

**Broader Impact Concerns:**

The groceries experiment touches price discrimination; reporting >100% revenue gains without fairness analysis could encourage harmful practices. A brief fairness/consumer‑welfare discussion and guardrails would strengthen the Broader Impact section.

**Claims And Evidence:**

No

**Claims Explanation:**

1. Eq. (5) uses an absolute difference over DR rewards; it should not yield negatives, yet Table 3 shows negative improvements (e.g., REBOA for some baselines). Moreover, Sec. 3.5 explicitly states that DR rewards on the test set lack natural meaning, but the Abstract/Contributions summarize results as reductions in complication/mortality rates, which over‑interprets the DR metric and invites misreading as real‑world rate reductions.
2. No dose‑grid sensitivity is given for continuous treatments (Sec. 2.3.2), and the cross‑entropy analogy in Sec. 2.4.1 lacks empirical stability diagnostics (e.g., temperature/entropy regularization ablations).

**Requested Changes:**

1. Negative values in Table 3 (REBOA) are inconsistent—either the formula or tables/implementation needs correction. Clarify sign conventions for minimization vs. maximization tasks.

2. Replace “reduction in complication/mortality rates by X%” with “DR test‑value improvement by X%” (or calibrate rewards to real units), consistent with Sec. 3.5’s note that DR rewards on test lack natural meaning.

3. Conduct paired tests (paired t or permutation) at the split level (N=5) against the next‑best method for each dataset; report p‑values/effect sizes.

4. Provide off‑policy evaluation (OPE) uncertainty for policy value (e.g., DR with bootstrap + cross‑fitting), especially for REBOA where overlap is weak.

5. The text embeddings sometimes come from a classification head fine‑tuned on outcomes. Provide audit of pre‑treatment time windows for notes and a sensitivity analysis that removes sentences containing explicit treatment/outcome tokens, to guard against label leakage.

6. Report dose‑grid sensitivity (step size / candidate set) for Diabetes and Groceries; if possible, compare against a continuous‑action baseline (e.g., policy gradient/actor‑critic).

---

> ### Author Response · Authors · 2025-12-14
> **Response to Reviewer g9eQ - Part 1**
>
> **1. Eq. (5) uses an absolute difference over DR rewards; it should not yield negatives, yet Table 3 shows negative improvements (e.g., REBOA for some baselines). Moreover, Sec. 3.5 explicitly states that DR rewards on the test set lack natural meaning, but the Abstract/Contributions summarize results as reductions in complication/mortality rates, which over‑interprets the DR metric and invites misreading as real‑world rate reductions.**
>
> Response: We thank the reviewer for the comment, we indeed mistakenly added the absolute value in the formula. The right formula, also updated in the paper, is:
>
> $$\bar{I} = \frac{ \sum_{i=1}^{n} ( \Gamma_{i,t_{i}} - \Gamma_{i,\hat{t}_{i}} )} {\sum_{i=1}^{n}\Gamma_{i,t_{i}}},$$
>
> for minimization problems. For maximization problems, the sign is the opposite. Regarding the second point, we agree with the reviewer that the previous version of the Abstract and Contributions overstated this point, and we have revised both sections to clearly indicate that all reported improvements concern estimated counterfactual outcomes.
>
> **2.a No dose‑grid sensitivity is given for continuous treatments (Sec. 2.3.2), and the cross‑entropy analogy in Sec. 2.4.1 lacks empirical stability diagnostics (e.g., temperature/entropy regularization ablations).**
>
> Response: We thank the reviewer for the comment. The diabetes raw dataset was provided to us in a discretized manner, with 12 distinct prescriptions possible. We use all 12 prescriptions as possible prescriptions for our models. We believe that these discretized prescriptions are true to real life, since, although possible to be continuous, medical prescriptions are realistically given by providers in discrete doses. The number of distinct prices in the groceries raw dataset we use includes: 1.99, 2, 2.49, 2.5, 2.99, 3.49, 3.5, 3.99. Given these prices, they are almost already discretized; we realistically round up the non-integral prices and achieve discretized prices of 2, 2.5, 3, 3.5, 4. With the nature of these datasets, we believe it may not be relevant to perform dose-grid sensitivity.
>
> **2.b The cross‑entropy analogy in Sec. 2.4.1 lacks empirical stability diagnostics (e.g., temperature/entropy regularization ablations).**
>
> Response: We thank the reviewer for the comment. To address the lack of empirical evidence on the convergence claims, we conducted two sets of experiments for different values of the temperature parameter τ and the entropy regularization weight λ. First, fixing λ=0, we varied the temperature τ; and second, fixing τ=1.0, we varied the regularization weight λ. Across all datasets, we observe smooth convergence behavior for both training and validation losses. The TAVR and Liver datasets display slightly more variability, which is expected given their smaller sample sizes and higher feature dimensionality. Convergence is robust for all values of τ, with larger temperatures leading to slower but more stable convergence. Varying λ has minimal effect on loss magnitude, except in the Groceries and Spleen datasets, where stronger entropy regularization (λ) slightly increases the converged loss value. Detailed plots can be found in the new Appendix Section A.1.
>
> **Requested Changes:**
>
> **a) Negative values in Table 3 (REBOA) are inconsistent—either the formula or tables/implementation needs correction. Clarify sign conventions for minimization vs. maximization tasks.**
>
> Response: We thank the reviewer for the comment, we indeed mistakenly added the absolute value in the formula. The right formula, also updated in the paper, is:
>
> $$\bar{I} = \frac{ \sum_{i=1}^{n} ( \Gamma_{i,t_{i}} - \Gamma_{i,\hat{t}_{i}} )} {\sum_{i=1}^{n}\Gamma_{i,t_{i}}},$$
>
> for minimization problems. For maximization problems, the sign is the opposite.
>
> **b) Replace “reduction in complication/mortality rates by X%” with “DR test‑value improvement by X%” (or calibrate rewards to real units), consistent with Sec. 3.5’s note that DR rewards on test lack natural meaning.**
>
> Response: Thank you for the suggestion. We have updated the wording in the abstract, the captions of Tables 1, 3, 11, 13, 16, and our contributions (Section 1.2). Please note that in these table captions,  we have chosen the wording “estimated outcome rewards” or “estimated outcome rewards (test set)”  to indicate that the values are rewards rather than actual outcome values, and we use “estimated” to capture both the direct method of reward estimation for the groceries dataset and the doubly robust estimation method used for the other datasets. We have used “estimated outcome rewards” in all other mentions of improvement throughout the paper.

---

> ### Author Response · Authors · 2025-12-14
> **Response to Reviewer g9eQ - Part 2**
>
> **c) Conduct paired tests (paired t or permutation) at the split level (N=5) against the next‑best method for each dataset; report p‑values/effect sizes.**
>
> Response: We thank the reviewer for the comment. We conducted paired significance tests (paired t-tests and permutation tests) between top-performing and next-best methods across splits and datasets, taking into account the N=5 splits per split type (50/50, 60/40, 70/30 and 80/20). The paired significance tests confirm that, in most cases, our proposed models perform comparably to the next-best method, with a few instances showing statistically significant superiority according to paired t-tests, and that improvements are modest, but consistent across the different datasets. Because each comparison is conducted over only five random splits, both the paired t-test and the permutation test are conservative. The full results are added in Table 14 of the Appendix Section A.2.2.
>
> **d) Provide off‑policy evaluation (OPE) uncertainty for policy value (e.g., DR with bootstrap + cross‑fitting), especially for REBOA where overlap is weak.**
>
> Response: Thank you for the suggestion. We have included the off-policy evaluation (OPE) uncertainty in the new Appendix Section A.4. Since we have trained a separate rewards model on the test set as discussed in the main text, we have used these estimated test set rewards to perform the bootstrapping procedure in this section of the appendix. These findings confirm that our models are comparable to or better than existing state-of-the-art prescriptive methods on structured datasets and are able to improve outcomes on unstructured datasets. In particular, the results on the TAVR dataset are quite stable; in the liver injury dataset, the wider confidence intervals are attributed to the small dataset size and the relatively smaller overlap between treatment groups. We observe that the groceries and diabetes datasets are the ones with the narrower confidence intervals. Although the REBOA dataset has the smallest positivity overlap, the results are quite robust, partially increasing our confidence in their interpretation.
>
> **e) The text embeddings sometimes come from a classification head fine‑tuned on outcomes. Provide audit of pre‑treatment time windows for notes and a sensitivity analysis that removes sentences containing explicit treatment/outcome tokens, to guard against label leakage.**
>
> Response: We thank the reviewer for the comment, as it is a valid concern. For both the TAVR and the liver trauma datasets, the notes come from medical reports that are written strictly prior to each of the datasets’ respective procedural treatments. Furthermore, we ensure that no textual content reveals future information by verifying that no treatment-related keywords (e.g. Evolut, Sapien in the TAVR dataset and the specific relevant surgery names in the liver trauma dataset) and outcome-related terms appear in the text. For TAVR, we search for the outcome terms pacemaker, implantation, PPI to confirm this. For liver trauma, since the outcome is patient mortality and the notes are written prior to patient treatment, the outcome is also not included in the notes. We also include an additional note-to-treatment time distribution plot for TAVR. The full discussion may be found in Appendix Section A.11.
>
> **f) Report dose‑grid sensitivity (step size / candidate set) for Diabetes and Groceries; if possible, compare against a continuous‑action baseline (e.g., policy gradient/actor‑critic).**
>
> Response: We thank the reviewer for the comment. The diabetes raw dataset was provided to us in a discretized manner, with 12 distinct prescriptions possible. We use all 12 prescriptions as possible prescriptions for our models. We believe that these discretized prescriptions are true to real life, since, although possible to be continuous, medical prescriptions are realistically given by providers in discrete doses. The number of distinct prices in the groceries raw dataset we use includes: 1.99, 2, 2.49, 2.5, 2.99, 3.49, 3.5, 3.99. Given these prices, they are already discretized; we realistically round up the non-integral prices and achieve discretized prices of 2, 2.5, 3, 3.5, 4. With the nature of these datasets, we believe it may not be relevant to perform dose-grid sensitivity or to train a continuous-action baseline model.
>
> **Broader Impact Concerns:**
> **The groceries experiment touches price discrimination; reporting >100% revenue gains without fairness analysis could encourage harmful practices. A brief fairness/consumer‑welfare discussion and guardrails would strengthen the Broader Impact section.**
>
> Response: Thank you for the comment. We have included a brief discussion on fairness and consumer welfare in the third paragraph of Section 4.3.

---

### Review · Reviewer_wgtL · 2025-11-19

**Summary Of Contributions:**

This paper introduces Prescriptive Neural Networks (PNNs) for treatment prescription using multimodal data, claiming to be the first prescriptive method handling multimodal inputs. They show improvements on medical datasets and comparable performance to existing methods on tabular data.

**Audience:**

Yes

**Audience Explanation:**

Policy learning is an interesting and challenging problem.

**Broader Impact Concerns:**

No.

**Claims And Evidence:**

No

**Claims Explanation:**

* All methods are evaluated on estimated counterfactuals rather than ground
truth, Models aligned with reward estimator biases appear better, but
this does not validate real-world performance. (I lack deep causal
inference expertise, but is relying entirely on estimated outcomes
without any validation against real data standard practice?)

* "First multimodal prescriptive method" seems like a stretch. Authors
  just concatenates pretrained embeddings with tabular features
  (standard practice). Core architecture is conventional feedforward
  network. I suggest a more humble wording.

* No evidence that SUTVA/ignorability/positivity hold. Post-hoc trimming arbitrary (5th/95th percentiles). REBOA has "exceptionally low" overlap but they proceed anyway

* Performance varies wildly by reward estimator (5% vs 21% improvement
in Table 1). Best embedding type selected per dataset without
justification. How robust are results to these choices? How would one
choose in practice?

* I also did not find authors' response to previous reviews particularly
convincing, e.g.,authors deflect the statistical significance issues by claiming "comparable" performance and cherry-pick specific splits showing significance, but most results in Tables 3, 9 still have overlapping error bars.

**Requested Changes:**

See my comments above (all critical)

---

> ### Author Response · Authors · 2025-12-14
> **Response to Reviewer wgtL Part 1**
>
> **1. All methods are evaluated on estimated counterfactuals rather than ground truth, Models aligned with reward estimator biases appear better, but this does not validate real-world performance. (I lack deep causal inference expertise, but is relying entirely on estimated outcomes without any validation against real data standard practice?)**
>
> Response: We thank the reviewer for this thoughtful comment. We agree that evaluating policies using estimated counterfactual outcomes does not reflect ground-truth clinical outcomes, which are fundamentally unobservable for treatments not historically assigned. However, doubly robust (DR) and related off-policy estimators are the accepted standard for policy evaluation in observational datasets, precisely because real-world counterfactuals cannot be observed. This practice is widely adopted in high-impact work, including clinical decision-making systems evaluated via off-policy estimation (e.g., Komorowski et al., Nature Medicine 2018; Amram et al., Machine Learning 2022; Louizos et al., NeurIPS 2017).
>
> We now explicitly acknowledge this limitation in a newly added paragraph in Section 3.5. To ensure that our evaluation is as rigorous and unbiased as possible, we (1) prevent any data leakage by ensuring that the counterfactual model used for test-set evaluation is trained exclusively on held-out data, and (2) quantify uncertainty via bootstrap off-policy evaluation in the Appendix Section A.4, allowing us to report confidence intervals around our estimated policy values.
>
> **2. "First multimodal prescriptive method" seems like a stretch. Authors just concatenates pretrained embeddings with tabular features (standard practice). Core architecture is conventional feedforward network. I suggest a more humble wording.**
>
> Response: We thank the reviewer for this helpful comment. We agree that the original phrasing could be softened, and we have revised it to more accurately reflect the contribution: our approach is among the first prescriptive methods evaluated on both structured and unstructured data within a unified framework. We have updated the abstract, the contributions section, and the conclusions accordingly.
>
> **3) No evidence that SUTVA/ignorability/positivity hold. Post-hoc trimming arbitrary (5th/95th percentiles). REBOA has "exceptionally low" overlap but they proceed anyway.**
>
> Response: We thank the reviewer for raising these important points. As in all observational causal inference and prescriptive learning settings, identification necessarily relies on the standard assumptions of SUTVA, conditional ignorability, and positivity. These assumptions are required for counterfactual estimation and are shared by all baseline methods evaluated in the paper.
>
> Regarding trimming, prior work (e.g., Stürmer et al., 2021) shows that percentile-based trimming often achieves superior bias–variance tradeoffs compared with fixed cutoffs, particularly in multi-treatment settings. In preliminary experiments, fixed thresholds such as [0.1,0.9] were overly conservative and challenging to tune across multiple treatment groups, frequently discarding more data than necessary. To preserve sample size while still enforcing overlap, we adopt a percentile-based strategy inspired by this literature, retaining 90% of each treatment group by removing the lowest and highest 5% of propensity scores (below the 5th percentile or above the 95th percentile), an approach already tested in (Stürmer et al. (2010); Glynn et al. (2019); Stürmer et al. (2021)). We have revised the manuscript to clarify this rationale in Section 4.1.1.
>
> We also note that many applied studies, including work using datasets similar to REBOA (e.g., Amram et al., Machine Learning 2022), do not apply trimming at all and rely solely on doubly-robust or IPW estimators. Our procedure therefore serves as an additional, conservative safeguard rather than a minimal requirement.
>
> For REBOA, where overlap remains limited even after trimming, we interpret the estimates with appropriate caution and quantify uncertainty via bootstrap off-policy evaluation. The resulting confidence intervals, presented in the Appendix Section A.4,  are not excessively wide, suggesting that despite limited overlap, the estimates remain informative.

---

> > ### Author Response · Authors · 2025-12-14
> > **Response to Reviewer wgtL Part 2**
> >
> > **4) Performance varies wildly by reward estimator (5% vs 21% improvement in Table 1). Best embedding type selected per dataset without justification. How robust are results to these choices? How would one choose in practice?**
> >
> > Response: We thank the reviewer for these thoughtful observations. The variation in absolute improvement (e.g., 5% vs. 21%) reflects differences in the reward estimators used for off-policy evaluation. Given the known variability across counterfactual outcome models, we not only train separate reward models for each modality (tabular vs. multimodal), but also evaluate performance when models from multiple estimator families (Doubly Robust, TARNet, Dragonnet) are used for training, as reported in Appendix Table 16. Across all these settings, the direction of the effect is consistent: multimodal PNNs outperform the tabular baselines.
> > Regarding embedding choices, we actually report results for all three embedding variants (full, classification-head, PCA) across datasets, splits, and reward estimators in Appendix Section A.2.1. The main text highlights the embedding achieving the best validation performance for clarity, reflecting the standard model-selection procedure one would adopt in practice. Importantly, the observed multimodal improvements remain robust across all embedding types.
> >
> > **5) I also did not find authors' response to previous reviews particularly convincing, e.g.,authors deflect the statistical significance issues by claiming "comparable" performance and cherry-pick specific splits showing significance, but most results in Tables 3, 9 still have overlapping error bars.**
> >
> > Response: We thank the reviewer for the comment. To address statistical significance properly, we conducted paired t-tests and paired permutation tests between the top-performing and next-best methods across all structured datasets and split types (N=5 per split type). As reported in Appendix Table 14, these tests show that the proposed models are always among the top 2 contestants and also statistically comparable to the next-best method in most settings, with some cases showing significant improvements. Given the small number of splits, both tests are conservative, which naturally limits the strength of detectable differences even when trends consistently favor our approach.
> > In addition, we now include off-policy evaluation (OPE) uncertainty estimates using DR with bootstrap, with special attention to REBOA where overlap is weaker (Appendix Section A.4). These additions provide a more complete assessment of variability in estimated policy values. We have clarified these points in the manuscript.

---

### Review · Reviewer_j5ZD · 2025-12-01

**Summary Of Contributions:**

This paper introduces Prescriptive Neural Networks (PNNs), a deep learning framework for optimal treatment prescription that can handle multimodal data. The authors evaluate PNNs on two multimodal medical datasets (TAVR and liver trauma) and four unimodal tabular datasets. The key innovation is the ability to process both structured and unstructured data (clinical notes, imaging reports) through embeddings, combined with counterfactual estimation to prescribe outcome-optimizing treatments. For interpretability on tabular data, the authors introduce Mirrored Optimal Classification Trees (Mirrored OCTs) that distill PNN prescriptions into transparent decision trees.

**Audience:**

Yes

**Audience Explanation:**

This is a paper on a very relevant topic of interest to the community

**Claims And Evidence:**

Yes

**Claims Explanation:**

The authors have made a significant improvement in the paper. For the sake of completeness a summary of the key claims are as below:
1. To my knowledge, this is indeed the first prescriptive method designed to handle multimodal data, which is highly relevant for real-world healthcare applications where decisions must integrate structured data with clinical notes and imaging reports.
2. The paper demonstrates the framework across six diverse real-world datasets spanning multiple treatment scenarios (binary, discrete multiple, single continuous, and multiple continuous treatments), showing versatility and broad applicability.
3. The revision has substantially strengthened the causal inference methodology by explicitly stating assumptions (SUTVA, ignorability, positivity), implementing propensity score trimming with clear diagnostic criteria, testing multiple counterfactual estimation methods (doubly robust, TARNet, Dragonnet, Causal Forests) and providing extensive appendices with overlap diagnostics and covariate balance assessments

4. As before, the Mirrored OCT approach successfully recovers interpretability on tabular datasets with minimal performance degradation, which is critical for clinical deployment. In addition, the paper includes comprehensive analyses of prescription realism, stability across splits, and treatment space coverage, addressing practical deployment concerns.

5. Finally, PNNs demonstrate statistically significant improvements on the groceries dataset and comparable performance to state-of-the-art methods on other datasets, while being the only method capable of handling multimodal data.

Given these tremendous effort, there are still a few aspects that might need further insights
1. The question of statistical significance remains partially unresolved. The additional splits reported in the appendix help, but the reported results for 50-50 splits are still somewhat overlapping
2. The authors added a convergence property - however, the paper might still benefit from formal proofs around theoretical bounds for approximation error and analysis of when the softmax indicator is appropriate.
3. In addition some of the additions such as the REBOVA dataset (see below) and trimming strategies may need further clarifications.
4. Finally the computational cost aspect still remain under-appreciated in the paper

**Requested Changes:**

Some changes that can improve the paper are as follows:

1. Question for the author - do you consider the possibility of positivity violations in REBOVA dataset and should this be removed from the main paper or additional caveats are provided? Further the trimming analysis is somewhat arbitrary and the impact on overlap after trimming may need more clarifications

2. Please provide a more detailed computational cost analysis

---

> ### Author Response · Authors · 2025-12-14
> **Response to Reviewer j5ZD Part 1**
>
> **1. The question of statistical significance remains partially unresolved. The additional splits reported in the appendix help, but the reported results for 50-50 splits are still somewhat overlapping.**
>
> Response: We thank the reviewer for the comment. To address statistical significance properly, we conducted paired t-tests and paired permutation tests between the top-performing and next-best methods across all datasets and split types (N=5 per split type). As reported in Appendix Table 14, these tests show that the proposed models are always among the top 2 contestants and also statistically comparable to the next-best method in most settings, with some cases showing significant improvements. Given the small number of splits, both tests are conservative, which naturally limits the strength of detectable differences even when trends consistently favor our approach.
> In addition, we now include off-policy evaluation (OPE) uncertainty estimates using DR with bootstrap, with special attention to REBOA where overlap is weaker (Appendix Section A.4). These additions provide a more complete assessment of variability in estimated policy values.
>
> **2. The authors added a convergence property - however, the paper might still benefit from formal proofs around theoretical bounds for approximation error and analysis of when the softmax indicator is appropriate.**
>
> Response: We thank the reviewer for the comment. To address the lack of evidence on the convergence claims, we conducted two sets of experiments for different values of the temperature parameter τ and the entropy regularization weight λ. First, fixing λ=0, we varied the temperature τ; and second, fixing τ=1.0, we varied the regularization weight λ. Across all datasets, we observe smooth convergence behavior for both training and validation losses. The TAVR and Liver datasets display slightly more variability, which is expected given their smaller sample sizes and higher feature dimensionality. Convergence is robust for all values of τ, with larger temperatures leading to slower but more stable convergence. Varying λ has minimal effect on loss magnitude, except in the Groceries and Spleen datasets, where stronger entropy regularization (λ) slightly increases the converged loss value. Detailed plots can be found in the new Appendix Section A.1.
>
> **3. In addition some of the additions such as the REBOA dataset (see below) and trimming strategies may need further clarifications.**
>
> Response: We thank the reviewer for this thoughtful observation. We agree that the REBOA dataset shows limited covariate overlap, and we already note in the manuscript that estimates in this setting should be interpreted with caution. In light of this, we chose to retain REBOA in the main text because low-positivity scenarios commonly arise in clinical practice, and illustrating model behavior under such conditions can still be informative as long as the limitations are clearly stated. In addition, the confidence intervals obtained from our bootstrapping off-policy evaluation (Appendix Section A.4) are not prohibitively wide, suggesting that, even with limited overlap, the resulting estimates retain a meaningful signal.
>
> Regarding trimming, prior work (e.g., Stürmer et al., 2021) shows that percentile-based trimming often achieves superior bias–variance tradeoffs compared with fixed cutoffs, particularly in multi-treatment settings. In preliminary experiments, fixed thresholds such as [0.1,0.9] were overly conservative and challenging to tune across multiple treatment groups, frequently discarding more data than necessary. To preserve sample size while still enforcing overlap, we adopt a percentile-based strategy inspired by this literature, retaining 90% of each treatment group by removing the lowest and highest 5% of propensity scores (below the 5th percentile or above the 95th percentile), an approach already tested in (Stürmer et al. (2010); Glynn et al. (2019); Stürmer et al. (2021)). We have revised the manuscript to clarify this rationale in Section 4.1.1.
>
> We also note that many applied studies, including those using similar datasets (e.g., Amram et al., Machine Learning 2022), rely on doubly robust or IPW estimators without any trimming. In comparison, our trimming step provides an additional, conservative safeguard that increases adherence to causal inference best practice.
>
> **4. Finally the computational cost aspect still remain under-appreciated in the paper.**
>
> Response: We thank the reviewer for the comment. We added the new Appendix Section A.6, where the average computational time per method and dataset is reported. In brief, Regress & Compare (XGBoost-based) is consistently the fastest method, while PNNs require longer training time due to gradient-descent optimization; tree-based prescriptive models (OPTs) and Mirrored OCTs fall in between.

---

> ### Author Response · Authors · 2025-12-14
> **Response to Reviewer j5ZD Part 2**
>
> **Requested Changes:**
> **Some changes that can improve the paper are as follows:**
>
> **a) Question for the author - do you consider the possibility of positivity violations in REBOVA dataset and should this be removed from the main paper or additional caveats are provided? Further the trimming analysis is somewhat arbitrary and the impact on overlap after trimming may need more clarifications.**
>
> Response: We thank the reviewer for this thoughtful observation. We agree that the REBOA dataset shows limited covariate overlap, and we already note in the manuscript that estimates in this setting should be interpreted with caution. In light of this, we chose to retain REBOA in the main text because low-positivity scenarios commonly arise in clinical practice, and illustrating model behavior under such conditions can still be informative as long as the limitations are clearly stated.  In addition, the confidence intervals obtained from our bootstrapping off-policy evaluation (Appendix Section A.4) are not prohibitively wide, suggesting that, even with limited overlap, the resulting estimates retain a meaningful signal.
>
> Regarding trimming, prior work (e.g., Stürmer et al., 2021) shows that percentile-based trimming often achieves superior bias–variance tradeoffs compared with fixed cutoffs, particularly in multi-treatment settings. In preliminary experiments, fixed thresholds such as [0.1,0.9] were overly conservative and challenging to tune across multiple treatment groups, frequently discarding more data than necessary. To preserve sample size while still enforcing overlap, we adopt a percentile-based strategy inspired by this literature, retaining 90% of each treatment group by removing the lowest and highest 5% of propensity scores (below the 5th percentile or above the 95th percentile), an approach already tested in (Stürmer et al. (2010); Glynn et al. (2019); Stürmer et al. (2021)). We have revised the manuscript to clarify this rationale in Section 4.1.1.
>
> We also note that many applied studies, including those using similar datasets (e.g., Amram et al., Machine Learning 2022), rely on doubly robust or IPW estimators without any trimming. In comparison, our trimming step provides an additional, conservative safeguard that increases adherence to causal inference best practice.
>
>
> **b) Please provide a more detailed computational cost analysis.**
>
> Response: We thank the reviewer for the comment. We added the new Appendix Section A.6, where the average computational time per method and dataset is reported. In brief, Regress & Compare (XGBoost-based) is consistently the fastest method, while PNNs require longer training time due to gradient-descent optimization; tree-based prescriptive models (OPTs) and Mirrored OCTs fall in between. However, we expect the difference between PNNs and tree methods to decrease, if one trained trees of larger depth.

---

### Decision · Action_Editor_g2FU · 2026-02-03

**Recommendation:** Accept with minor revision

**Audience:**

Yes

**Audience Explanation:**

The paper tackles multimodal prescription learning with a simple, scalable objective which will be appreciated by a wide range of TMLR audience. I see no issues here and all reviewers agree.

**Claims And Evidence:**

Yes

**Claims Explanation:**

This is the revised version of a previously rejected paper from TMLR and had 2 of the original 3 reviewers. As far as I could see, the authors had done a good job of modifying their original work to take the reviewers concerns into account and provided a far stronger version of the paper. The new reviews also had several comments for the authors and the major points of contention were as follows:

* The question of statistical significance. Several reviewers pointed out that the study lacked any form of statistical significance tests and thus paired t-test or permutation tests can be conducted to actually understand the significance of the presented results.

* All the reviewers had the question about the REBOA dataset. The main point was that REBOA has an exceptionally low overlap and thus an off‑policy evaluation (OPE) uncertainty for policy value should be provided for a better clarity and understanding.

After the rebuttal, the new reviewer had outstanding concerns that improvements may be driven by reward-estimator bias or fragile causal assumptions. Although true, I do agree with the authors that they have followed the so called literary protocol and that the doubly robust ( and related off-policy estimators are the accepted standard for policy evaluation in observational datasets.

1 other reviewer was leaning towwards accept but was unwilling to champion the paper. The 3rd reviewer did not have any recommendation posted even after several reminders and thus I had to make a judgement on their behalf. I see that the authors have provided sufficient answers to the reviewers point and I expect them to lean accept in the final evaluation.

I thus recommend acceptance of the paper and hope that the authors do take all the comments into account while preparing the final version of the paper.

---

> ### Author Response · Authors · 2026-02-19
>
> We thank the editor for the acceptance and appreciate the time taken in the reply. To address the remaining reviewers’ points, we would like to emphasize, as you also kindly mentioned in your comments, that we evaluate our experiments according to the well-accepted principles of Off-Policy Evaluation. Acknowledging the limitations of evaluating prescriptive problems, we use different reward estimators on the training and test set, we perform multiple splits/bootstrap analysis, and we use different models for the reward estimation to not attribute the performance gains on only the Doubly Robust estimator. We have made this more explicit in the paper, Section 3.5.
>
> We just submitted the camera ready version of the paper, as indicated.